# Programming of refractive functions

Md Sadman Sakib Rahman[1,2,3], Tianyi Gan [1,3], Mona Jarrahi [1,3] & Aydogan Ozcan [1,2,3] ✉

Snell's law dictates the phenomenon of light refraction at the interface between two media. Here, we demonstrate arbitrary programming of light refraction through an engineered material where the direction of the output wave can be set independently for different directions of the input wave, covering arbitrarily selected permutations of light refraction between the input and output apertures. Formed by a set of cascaded transmissive layers with optimized phase profiles, this refractive function generator (RFG) spans only a few tens of wavelengths in the axial direction. In addition to monochrome RFG designs, we also report wavelength-multiplexed refractive functions, where a distinct refractive function is implemented at each wavelength through the same engineered material volume, i.e., the permutation of light refraction is switched from one desired function to another function by changing the illumination wavelength. As experimental proofs of concept, we demonstrate permutation and negative refractive functions at the terahertz part of the spectrum using 3D-printed materials. Arbitrary programming of refractive functions enables new design capabilities for optical materials, devices and systems.

The study of refraction dates back to ancient times when early philosophers like Ptolemy explored the bending of light as it passed through different media. This bending is dictated by the Snell's law, i.e., $n_{in} \sin \theta_{in} = n_{out} \sin \theta_{out}$, where $n_{in}$ and $n_{out}$ are the refractive indices of the two media, and $\theta_{in}$ and $\theta_{out}$ refer to the angles that the light rays have with respect to the surface normal, while the azimuthal angles of the incident and refracted rays remain the same, i.e., $\varphi_{in} = \varphi_{out}$[1,2]. Advances in nanotechnology have enabled engineering of artificial materials[3–11] with negative effective refractive indices[12–18], causing light to bend in unusual ways and giving rise to phenomena such as anomalous refraction[12,19–24] and perfect lensing[25–29]. However, the refractive function that relates the direction of the refracted wave $(\theta_{out}, \varphi_{out})$ to the direction of the incident wave $(\theta_{in}, \varphi_{in})$ is a fixed function determined by the refractive indices of the two media, as described by the Snell's law. This behavior arises from the phase-matching condition[30] of the wavefronts on both sides of an interface. Recognizing this fact allows the tuning of the refractive function along an interface by introducing a phase-gradient, leading to the generalized Snell's law, $n_{out} \sin \theta_{out} - n_{in} \sin \theta_{in} = \frac{\lambda}{2\pi} \frac{d\psi}{dx}$ where $\psi$ is the spatial phase distribution at the interface and $\lambda$ is the wavelength of light[31–35].

Such a phase-gradient can be implemented by using, e.g., gradient metasurfaces, where the properties of the subwavelength inclusions constituting the meta-atoms vary gradually across a surface[36–43]. While there are reports of tunable refraction realized by adjusting the optical properties of these engineered materials through external stimuli[44–50], at any given state of these materials, the behavior of light refraction for different input directions remains coupled. This prohibits arbitrary programming of the output wave direction independently for each input direction of light; as a result, arbitrary programming of refractive functions could not be achieved with these earlier designs.

Here we demonstrate arbitrary programming of refractive functions through a passive optical device, which we term refractive function generator (RFG); see Fig. 1. In an RFG, the independently optimizable spatial features, i.e., the discrete phase elements, are distributed at a lateral pitch of ~$\lambda/2$ over consecutive transmissive layers, axially spanning only ~$15\lambda - 50\lambda$. Supervised deep learning[51] is used to optimize the collection of these transmissive layers for the implementation of a desired two-dimensional refractive function ($f$), where $\hat{k}_{out} = f(\hat{k}_{in})$ and $\hat{k}_{in}, \hat{k}_{out}$ define the propagation directions of the input and output waves, respectively. We report RFG designs that can

[1]Electrical and Computer Engineering Department, University of California, Los Angeles, CA, USA. [2]Bioengineering Department, University of California, Los Angeles, CA, USA. [3]California NanoSystems Institute (CNSI), University of California, Los Angeles, CA, USA. ✉e-mail: ozcan@ucla.edu

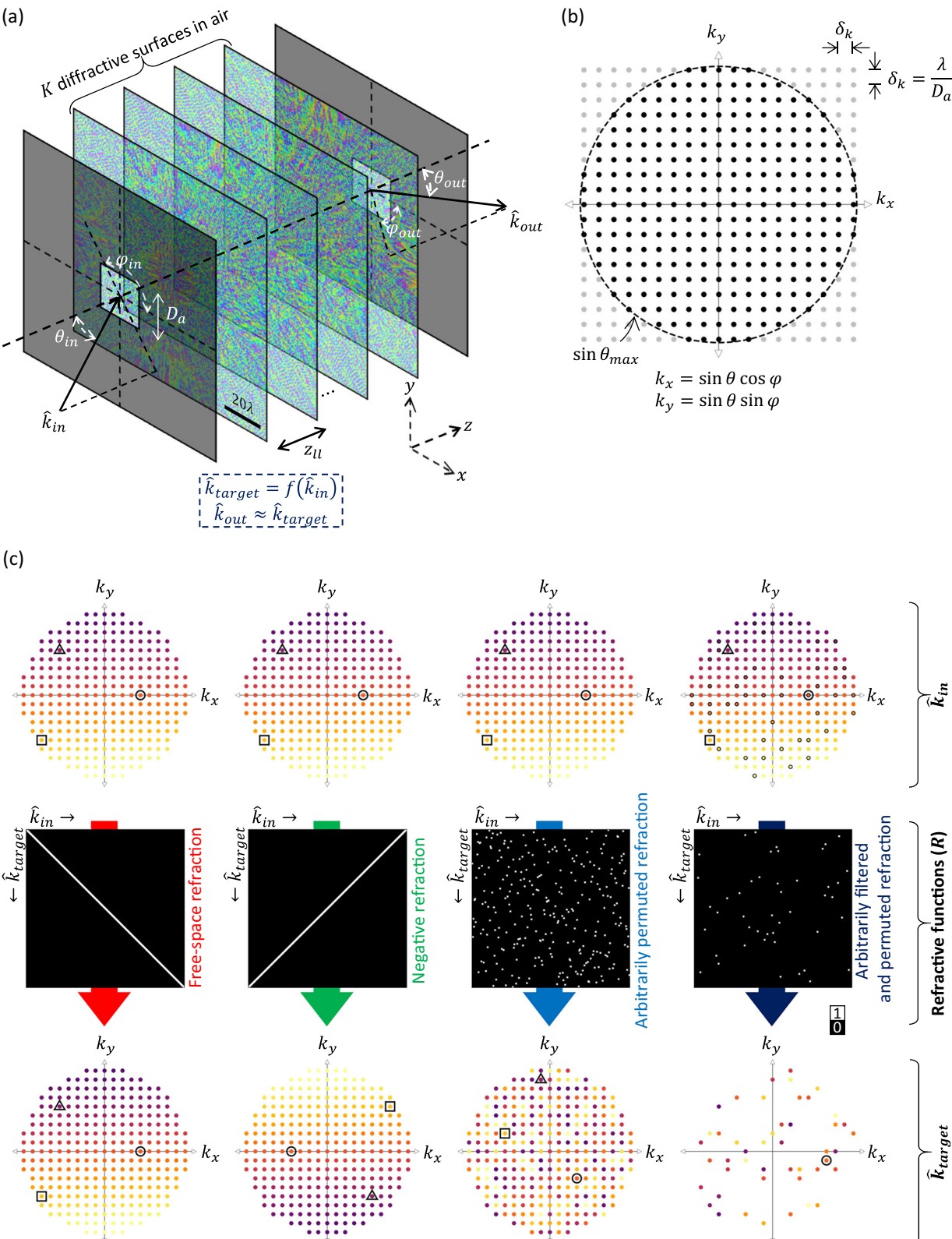

**Fig. 1 | Programming refractive functions. a** An RFG comprising $K$ transmissive surfaces in air with an axial separation of $z_{ll}$ between the successive surfaces (e.g., $z_{ll} \sim 6\lambda$). The RFG refracts an input wave along the direction $\hat{k}_{in}$ into the direction $\hat{k}_{out}$, where $\hat{k}_{out} \approx \hat{k}_{target} = f(\hat{k}_{in})$ and $f$ is the target refractive function of interest. **b** The set of all $\hat{k}$ vectors of interest for a given maximum angle $\theta_{max}$ and a finite aperture width of $D_a$, represented by the dots. **c** The mapping of the $\hat{k}$ vectors under different refractive functions $f$ represented by the binary matrices $R$. The mapping is encoded in the color of the dots. For visual aid, the mappings of three $\hat{k}$ vectors are also highlighted with a triangle, a circle, and a square to guide the eye.

achieve an arbitrary mapping between the directions of the input and output waves, i.e., for any given direction of input light, the output follows an arbitrarily selected direction for the refracted light, covering any desired permutation function between the input light and the refracted output light. Once the supervised optimization is complete for a given target RFG, the resulting design is fabricated and assembled to form the physical 3D material to perform the desired refractive function between the input and output waves passing through a thin optical volume. In addition to monochrome RFG designs, we also report the use of wavelength multiplexing to simultaneously execute a group of arbitrary refractive functions through the same thin material, each unique function performed at a separate wavelength. In these wavelength-multiplexed RFG designs, switching the illumination wavelength changes the refractive function, covering a set of independent mappings between the directions of the input and output waves. To show the proof of concept of an RFG design, we experimentally demonstrated the programming of a permutation refractive function and negative refractive function ($\theta_{out} = \theta_{in}$, $\varphi_{out} = \varphi_{in} + 180°$) at the terahertz (THz) part of the spectrum using 3D-printed devices. Without the need for dispersion engineering or deeply sub-wavelength material structures, arbitrary programming of refractive functions opens up new opportunities for the design of advanced optical devices and systems.

## Results

### Design and architecture of an arbitrary refractive function generator (RFG)

The architecture of an RFG is shown in Fig. 1a. A set of $K$ optimized transmissive surfaces, placed between the input and output apertures, form the core of an RFG. For this work, we only consider phase-only surfaces that modulate the phase of the incident wave. Without loss of generality, the amplitude modulation is assumed to be negligible, which is a valid assumption here, considering the short axial thickness of an RFG design. The RFG redirects a given input wave, propagating along the direction defined by the unit vector $\hat{k}_{in}$, into the output direction $\hat{k}_{out}$ such that $\hat{k}_{out} \approx \hat{k}_{target}$, where $\hat{k}_{target} = f(\hat{k}_{in})$ and $f$ is the target/desired refractive function, defining the mapping between the input and output directions. The unit vector $\hat{k}$ denotes the direction of the wavevector $\vec{k}$ of a plane wave, i.e., $\vec{k} = \frac{2\pi}{\lambda}\hat{k}$, where $\lambda$ is the wavelength of light. The unit vector $\hat{k}$ encapsulates the two angles $\theta$ and $\varphi$, i.e., polar/zenith angle and azimuthal angle[52] in a spherical coordinate system (see Fig. 1a), describing the propagation direction as follows:

$$\hat{k} = \begin{bmatrix} k_x \\ k_y \\ k_z \end{bmatrix} = \begin{bmatrix} \sin\theta\cos\varphi \\ \sin\theta\sin\varphi \\ \cos\theta \end{bmatrix} \tag{1}$$

Since $k_z^2 = 1 - k_x^2 - k_y^2$, a more succinct representation of $\hat{k}$ is the 2D-vector $(k_x, k_y)$[53]. Although $\theta$ and $\varphi$ can be continuous in principle, the resolution $\delta_k$ of $k_x$ and $k_y$ allowed by a finite aperture of dimension $D_a$ is also finite[54], i.e., $\delta_k \approx \frac{\lambda}{D_a}$. Therefore, for a given acceptance angle $\theta_{max}$ (the maximum angle with respect to the $z$ axis), the set $\mathbb{K}$ of all $\hat{k}$ vectors of interest can be written as: $\mathbb{K} = \{(k_x, k_y) : k_x = p\frac{\lambda}{D_a}, k_y = q\frac{\lambda}{D_a}, k_x^2 + k_y^2 < \sin^2\theta_{max}\}$ where $p$ and $q$ are integers. The elements of this set are represented by the dots within the circle of radius $\sin\theta_{max}$ in Fig. 1b, and can be enumerated as $\{\hat{k}_1, \hat{k}_2, \cdots, \hat{k}_{N_m}\}$ where $N_m = |\mathbb{K}|$ is the number of elements in $\mathbb{K}$. An arbitrary refractive function $f$ can be thought of as a mapping from $\mathbb{K}$ to $\mathbb{K}$, i.e., $f : \mathbb{K} \rightarrow \mathbb{K}$. Each of the dots, defining the set $\mathbb{K}$, represents a 'direction' that the input or output wave can have.

The mapping of $\mathbb{K}$ under an arbitrary refractive function $f$ can be described by a binary $N_m \times N_m$ matrix $R$ (such as the ones shown in Fig. 1c), where the 1's in the matrix define the coupling between $\hat{k}_{in}$ and $\hat{k}_{target}$. In other words, $R[p, q] = 1$ implies that if $\hat{k}_{in} = \hat{k}_q$, then $\hat{k}_{target} = f(\hat{k}_{in}) = \hat{k}_p$ where $p, q \in \{1, 2, \cdots, N_m\}$. We show a few examples of such matrices $R$ and the corresponding mappings of the input directions in Fig. 1c, where each mapping is encoded in the color of the elements of $\mathbb{K}$. For example, an identity matrix (first column of Fig. 1c) represents the free-space refractive function ($\hat{k}_{target} = \hat{k}_{in}$ or $\theta_{target} = \theta_{in}$, $\varphi_{target} = \varphi_{in}$), whereas the flipped identity matrix (second column of Fig. 1c) represents the negative refractive function ($\theta_{target} = \theta_{in}$, $\varphi_{target} = \varphi_{in} + 180°$). A more general form of an arbitrary refractive function can be represented by an arbitrarily selected permutation matrix, which defines an arbitrary mapping between $\hat{k}_{in}$ and $\hat{k}_{target}$ (third column of Fig. 1c). As another alternative for $f$, we can also envision an arbitrarily filtered and permuted refractive function (fourth column of Fig. 1c), where the input waves traveling in certain arbitrarily chosen directions (the ones corresponding to the columns with all zeros) are filtered out, whereas the waves in the other directions are redirected in a manner following the permutation defined by $f$ or the corresponding $R$.

The design of an RFG follows supervised learning using pairs of input direction $\hat{k}_{in}$ (equivalently, $\theta_{in}$, $\varphi_{in}$) and target direction $\hat{k}_{target}$ (equivalently, $\theta_{target}$, $\varphi_{target}$), defined based on the desired/target refractive function $f$ represented by $R$. This involves angular spectrum-approach based numerical simulation of wave propagation through a digital model of the RFG (see the "Methods" section). For a wavefront corresponding to $\hat{k}_{in}$ at the input aperture, the wavefront leaving the output aperture is numerically simulated; the error between the output wavefront and the wavefront corresponding to $\hat{k}_{target}$ is back-propagated to update and iteratively optimize the surface phase features using a gradient descent-based algorithm; see the "Methods" section for details. Unless otherwise stated, we assumed an operating wavelength of $\lambda = 0.75$ mm for the results shown in the following sections. However, we emphasize that the presented conclusions hold for any wavelength of interest, as long as the dimensions are scaled proportionally to the illumination wavelength, $\lambda$.

The results and analyses presented in the subsections leading up to the "Experimental results" as well as the animations presented in the Supplementary Movies 1–4 are based on numerical simulations. Unless otherwise stated, each diffractive surface in these simulations consists of $200 \times 200$ independently optimized phase features/elements. Each phase element spans an area of $0.53\lambda \times 0.53\lambda$, resulting in a total surface width of $106\lambda$. For arbitrary permutation refractive functions, both the input and output apertures have a width of $10.6\lambda$; for negative refractive functions, this width is set to $15.9\lambda$. The separation between consecutive planes—whether they contain input/output apertures or diffractive surfaces—is set to $6\lambda$, unless specified otherwise. The maximum input polar angle $\theta_{max}$ is assumed to be $60°$.

### Arbitrarily permuted refractive functions

We begin with the design of an arbitrarily permuted refractive function, where the target mapping between the input and output directions is defined by an arbitrarily chosen permutation matrix $R$ (see Fig. 2a and the 3rd column of Fig. 1c). We designed an RFG comprising $K = 8$ structured surfaces to implement this refractive function, where the axial distance between two consecutive surfaces $z_{ll}$ was $6\lambda$, giving an axial span of $z_{1K} \approx 50\lambda$ between the first and the last layers. The optimized phase profiles of these surfaces are shown in Fig. 2c. For each input direction $\hat{k}_{in}$, the corresponding output angle error is also shown in Fig. 2b. This output angle error $\varepsilon$ is defined as the angle between $\hat{k}_{out}$ and $\hat{k}_{target} = f(\hat{k}_{in})$. The estimation of the output wave direction $\hat{k}_{out}$ from the output wavefront is described in the "Method" section. Figure 2b reveals that the angular errors between the output

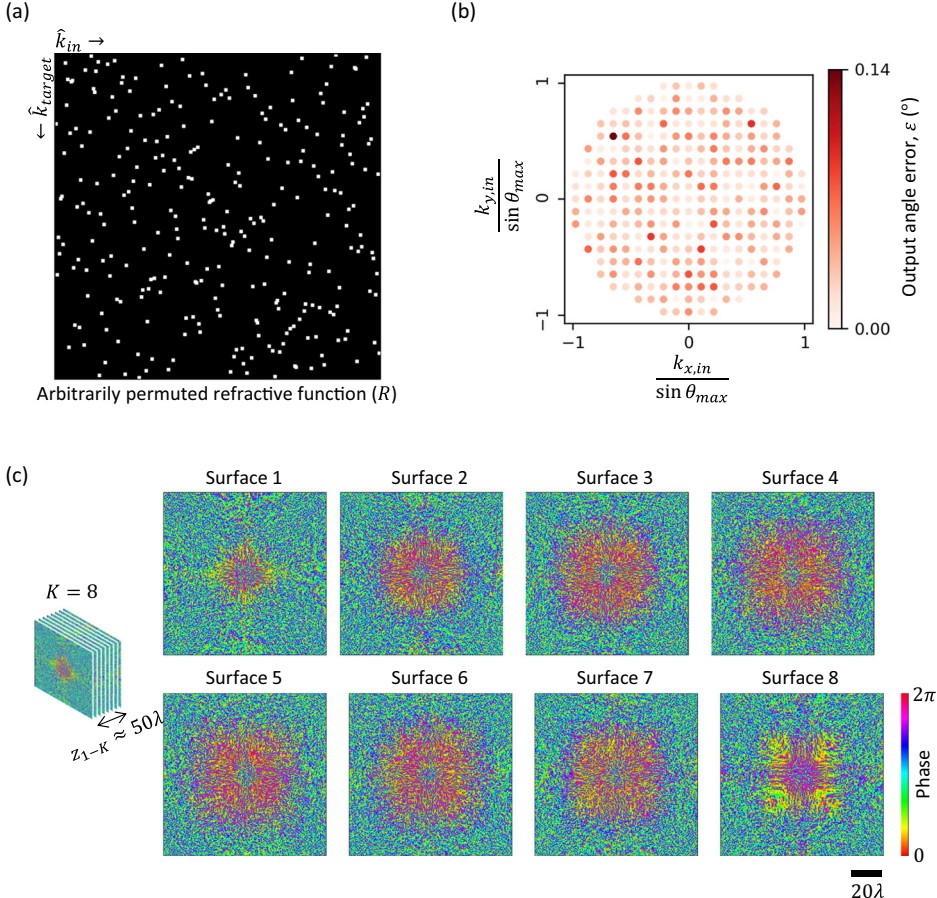

(a)
$\hat{k}_{in} \rightarrow$
$\leftarrow \hat{k}_{target}$
Arbitrarily permuted refractive function ($R$)

(b)
$\frac{k_{y,in}}{\sin \theta_{max}}$
$\frac{k_{x,in}}{\sin \theta_{max}}$
Output angle error, $\varepsilon$ (°)

(c)
$K = 8$
$z_{1-K} \approx 50\lambda$
Surface 1   Surface 2   Surface 3   Surface 4
Surface 5   Surface 6   Surface 7   Surface 8
$2\pi$
Phase
0
$20\lambda$

**Fig. 2 | Arbitrarily permuted refractive function implementation with a $K = 8$ RFG design. a** The matrix $R$ representing the arbitrarily permuted refractive function, the same as the one depicted in Fig. 1c, 3rd column. **b** The error in output angles for all the input directions. **c** The optimized phase profiles of the RFG surfaces. Here $z_{ll} \approx 6\lambda$, giving a total axial thickness of $z_{1-K} \approx 50\lambda$ between the first and the last surfaces.

directions and the target directions are negligible (less than 0.14°), revealing the success of the RFG in implementing the arbitrarily permuted refractive function, i.e., $\hat{k}_{out} \approx \hat{k}_{target} = f(\hat{k}_{in})$. Supplementary Movie 1 also shows the far-field output intensity as the input wave direction is swept, together with the corresponding target patterns that follow $f$.

In Fig. 3, we further analyze the errors of the refractive function implementation as the wavelength $\lambda_{test}$ of the input light deviates from the design wavelength $\lambda_{train}$ that the RFG is trained to operate at. While evaluating the error as a function of $\lambda_{test}$, we kept the input and target directions at $\lambda_{test}$ the same as those at $\lambda_{train}$. At each test wavelength $\lambda_{test}$, the distribution of the angular errors (over $N_m$ different input directions) is encapsulated with a box-and-whisker diagram in Fig. 3a. As expected, the error increases as $\lambda_{test}$ deviates from $\lambda_{train}$. However, the angular errors remain below 4° over a wavelength range of $\sim$ $0.99\lambda_{train}$ to $\sim 1.01\lambda_{train}$. The resilience against changes in wavelength can be improved by incorporating random wavelength variations during training. In Fig. 3b, we report the angular error of another design which was trained by randomly selecting the illumination wavelength from a desired range of interest during training, i.e., $\lambda_{train} \sim$ Uniform(742.5 µm,757.5 µm). The angular errors of this "vaccinated" design remain limited to $\sim 1°$ over the same wavelength range, showing the flexibility of our design approach to adapt to different requirements.

Figure 3 further depicts the dependence of the output errors on the number of structured surfaces $K$ and the surface-to-surface

distance $z_{ll}$ comprising the RFG structure. For Fig. 3c, we decreased $K$ from 8 to 3, keeping $z_{ll} = 6\lambda$. The output angle errors increased as $K$ decreased; however, we can see that the errors remain below 1° even when $K$ is decreased to 4. For Fig. 3d, we set the number of structured surfaces $K = 4$ and reduced the surface-to-surface distance $z_{ll}$ from $6\lambda$ to $4\lambda$. On average, the output error increased with a decrease in $z_{ll}$. However, even with $K = 4$ and $z_{ll} = 4\lambda$, the output angle errors stay below 1.6°, demonstrating an arbitrarily permuted refractive function with an RFG spanning only $\sim 15\lambda$ along the axial direction. To clarify, each $K$ ($z_{ll}$) value in Fig. 3c (3 d) represents a separately trained RFG design for the same target refractive function as in Fig. 2a.

An important metric of an RFG design is the output diffraction efficiency (DE), i.e., ratio between the diffracted output power along the target direction and the incident power at the input aperture; see the "Methods" section. For the RFG reported in Fig. 2, the diffraction efficiencies along the target directions ranged from 0.07% to 0.62%, see the 1st row of Fig. 4a. We can tune the diffraction efficiency of an RFG design by properly modifying the training loss function. By using an additional term in the loss function, weighted by $\eta$ (a training hyperparameter), which penalizes against low diffraction efficiency, we can improve the output diffraction efficiencies of the resulting design with a relatively small sacrifice in the output error performance. For example, by using $\eta = 30$, we can have an RFG design where the maximum output angle error is 0.87°, while the minimum diffraction efficiency increases to 2.75% (see the 3rd row of Fig. 4a). Figure 4b

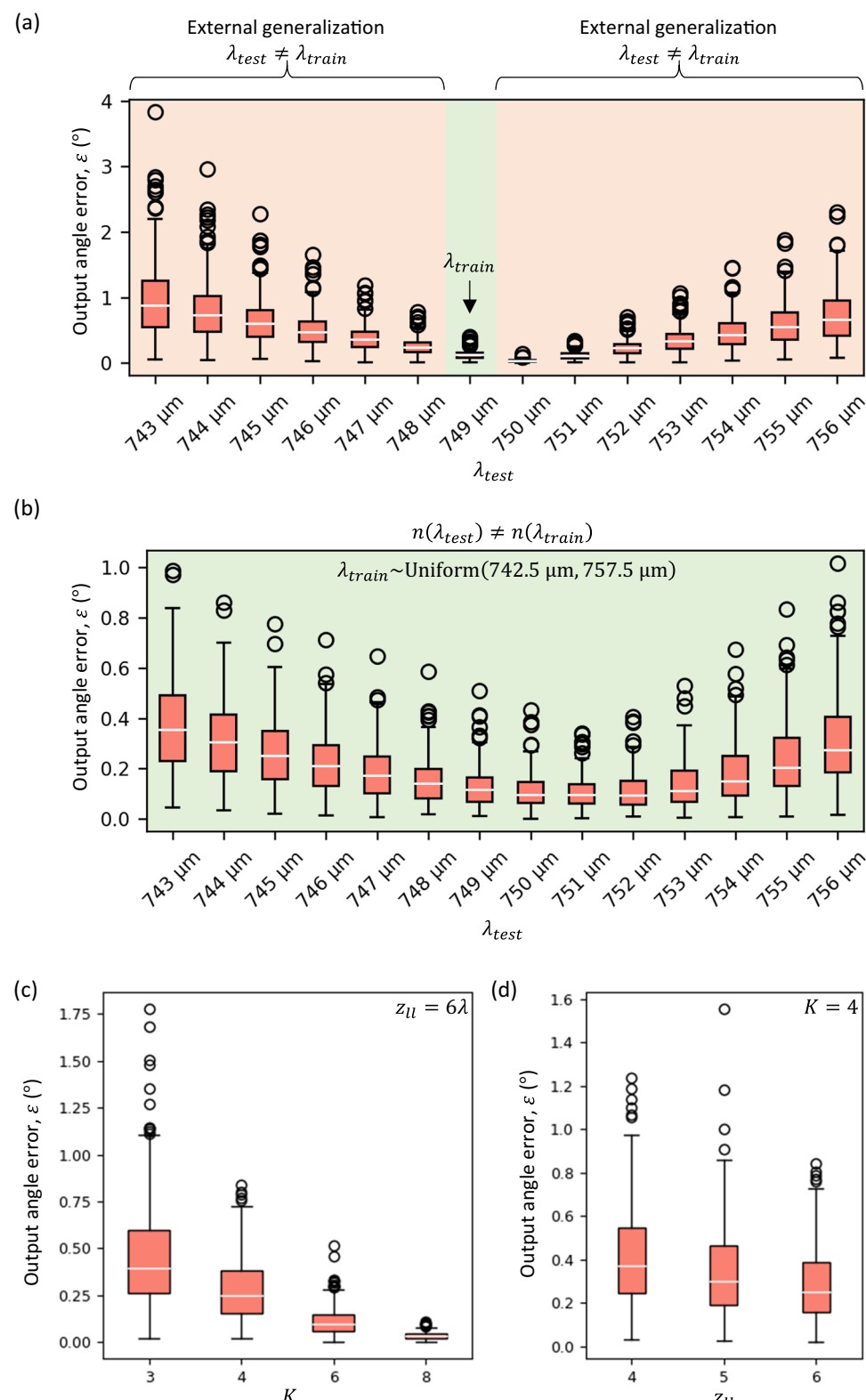

**Fig. 3 | Wavelength sensitivity and compactness of RFGs. a** Distribution of the output angle error as a function of the test wavelength $\lambda_{test}$, for the same RFG of Fig. 2, which was trained for an illumination wavelength of 750 μm, i.e., $\lambda_{train} = 750$ μm. The distributions arise from the values corresponding to all the input directions. **b** Distribution of the output angle error as a function of the test wavelength $\lambda_{test}$, for an RFG design "vaccinated" against changes in wavelength. **c** Distribution of the output angle errors as a function of $K$, while $z_{ll}$ is kept constant at $\sim 6\lambda$. **d** Distribution of the output angle errors as a function of $z_{ll}$, while $K$ is kept constant at 4.

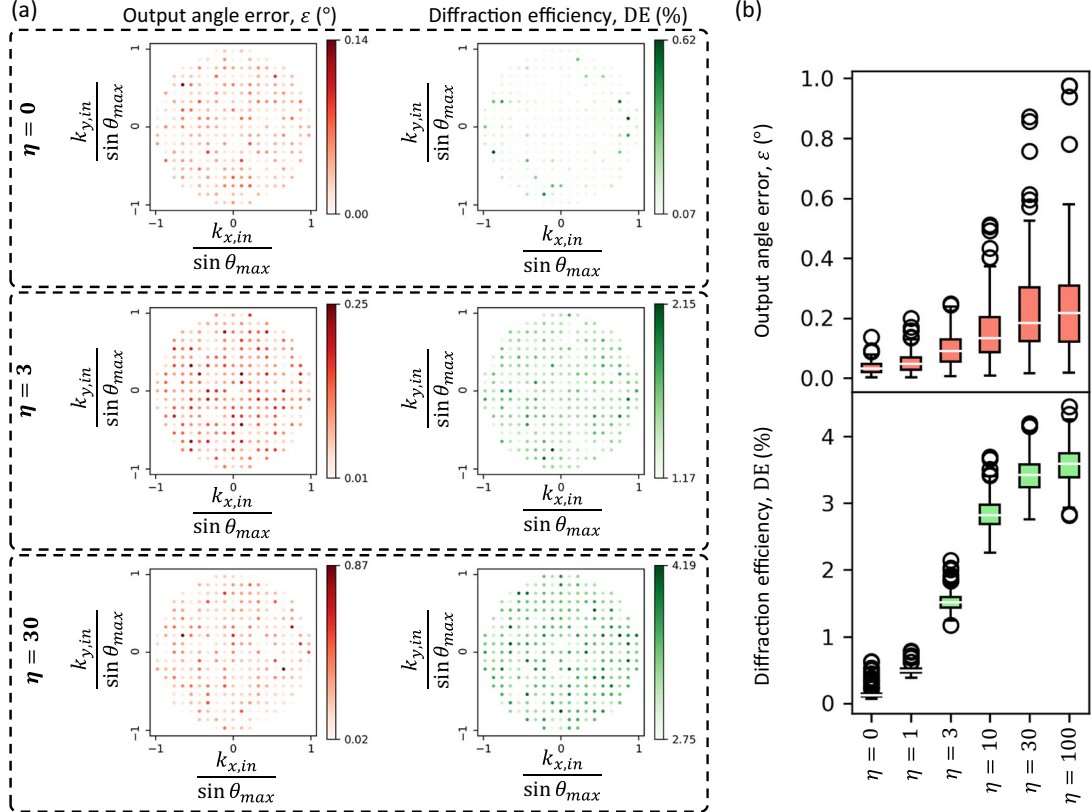

**Fig. 4 | Enhancement of the output diffraction efficiency of an RFG designed for an arbitrarily permuted refractive function. a** Output angle errors and diffraction efficiencies of three different RFG designs trained with different values of the hyperparameter $\eta$ (see Eq. (12)). Here, the target refractive function is the one shown in Fig. 2a. **b** Distribution of output angle errors and diffraction efficiencies as a function of $\eta$. Each value of the training hyperparameter $\eta$ corresponds to a separately optimized RFG design. For all the designs, $K = 8$ and $z_{ll} \approx 6\lambda$.

further summarizes the trade-off between the RFG performance and the output diffraction efficiency as a function of $\eta$.

### Arbitrarily filtered and permuted refractive functions

Next, we demonstrate the case of an arbitrarily filtered and permuted refractive function. Figure 5a depicts the target refractive function in this case. Here, ~90% of the input directions are filtered out at the output aperture, whereas the rest are redirected as specified by the non-zero elements of $R$; stated differently, $R$ in this case refers to an arbitrary permutation matrix with ~90% of its columns replaced with zeros, corresponding to the filtering of specific directions of input light. To implement this filtered refractive function, we designed an RFG comprising $K = 8$ surfaces, where the distance between consecutive surfaces $z_{ll}$ was $6\lambda$, yielding an axial span of $z_{1K} \approx 50\lambda$ between the first and the last surfaces. The optimized phase profiles of these surfaces are shown in Fig. 5e. Figure 5b reveals negligible errors in the output angles for the input directions which are not filtered. At the same time, the diffraction efficiencies along the targeted output directions are >10%; see Fig. 5c. To evaluate the filtering operation, we also calculated the relative percentage of residual power (i.e., the ratio between the power at the output and the power at the input aperture; see the "Methods" section) for each one of the filtered out directions. As shown in Fig. 5d, the relative power transmission is <1% for all the input directions to be filtered, correctly approximating this arbitrarily filtered and permuted refractive function.

### Negative refractive function

We also considered a specific form of refractive function, i.e., the negative refractive function, where $\theta_{target} = \theta_{in}$ and $\varphi_{target} = \varphi_{in} + 180°$;

see the third column of Figs. 1c and 6. To train for this refractive function, $\theta_{in}$ and $\varphi_{in}$ are randomly sampled from the uniform distributions Uniform$(0°, \theta_{max} = 60°)$ and Uniform$(0°, 360°)$, respectively. We designed an RFG comprising $K = 5$ surfaces for implementing the negative refractive function, and the optimized phase profiles are shown in Fig. 6c. For a dense grid of input directions $\hat{k}_{in}$, we show the corresponding output angle errors in Fig. 6a and the resulting diffraction efficiencies in Fig. 6b. While the maximum output angle error is ~2°, this relatively large error occurs only when $\theta_{in}$ is close to $\theta_{max} = 60°$ because of the limited amount of training examples around these angular values at the edges. Figure 6d depicts the "operating curve" of this RFG, which plots the maximum acceptable input angle $\theta_M$ vs. the maximum acceptable output angle error $\varepsilon_M$, such that $\varepsilon \le \varepsilon_M$ if $\theta_{in} \le \theta_M$. This plot shows that the RFG can operate at larger input angles if relatively larger errors are tolerated. For example, when $\theta_{in} \le 58°$, the output angle error stays below 1°, as shown in Fig. 6e. Also, Fig. 6b plots the output diffraction efficiency for all the input directions, revealing high diffraction efficiency even without the use of a diffraction efficiency-related term in the training loss function.

### Wavelength multiplexing of arbitrarily permuted refractive functions

Wavelength multiplexing can be used to implement completely different refractive functions, simultaneously executed through the same RFG with a unique refractive function assigned to each wavelength of interest. We demonstrated this wavelength multiplexing capability by designing an RFG that performs three different arbitrarily permuted refractive functions at three different wavelengths, as shown in Fig. 7a, top row. Without loss in generality, we chose the refractive functions

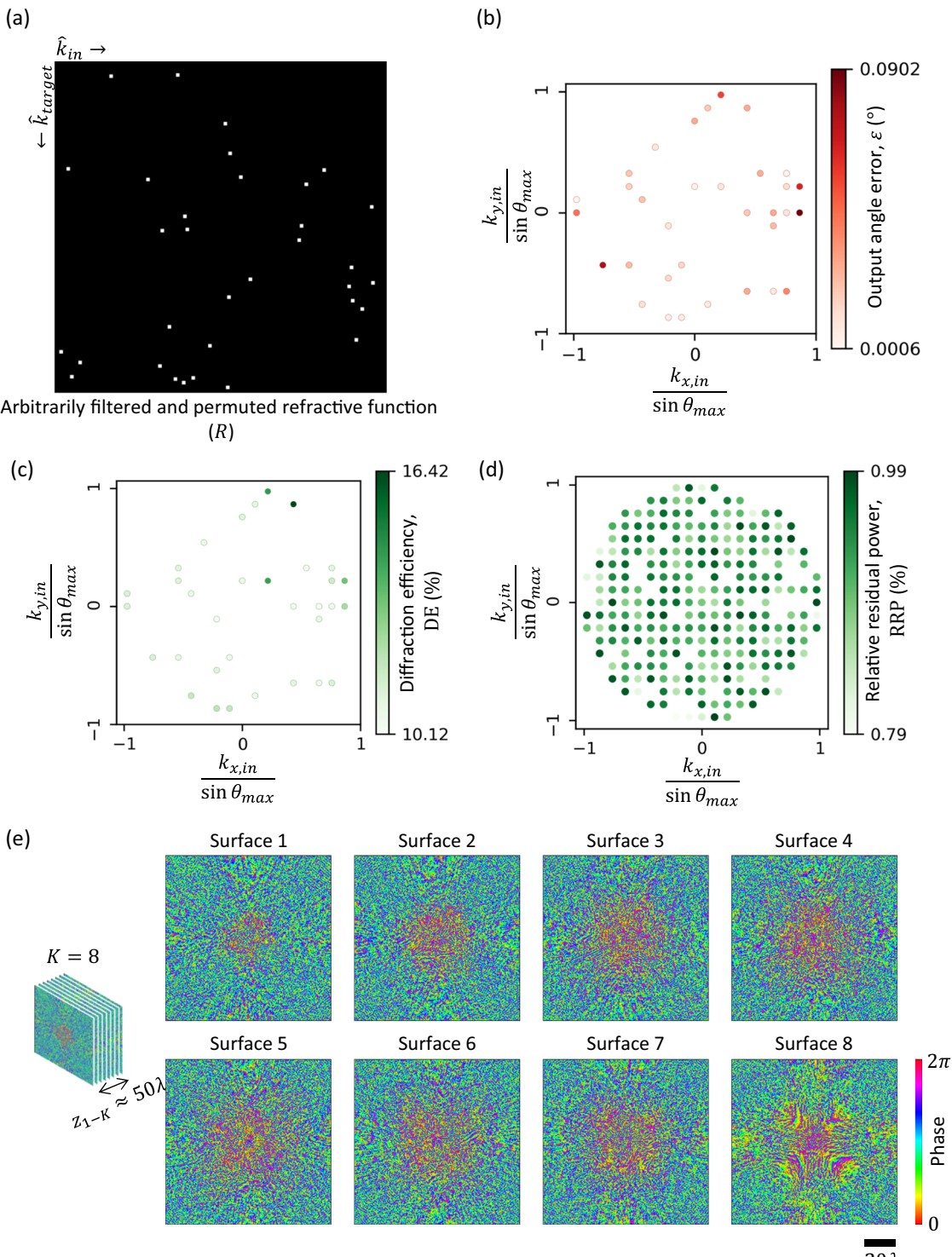

**Fig. 5 | Arbitrarily filtered and permuted refractive function implementation with a $K = 8$ RFG design. a** The matrix $R$ representing an arbitrarily filtered and permuted refractive function. The ratio of the filtered directions is -90%. **b** Output angle error $\varepsilon$ for all the unfiltered input directions. **c** Output diffraction efficiency DE for all the unfiltered input directions. **d** Relative residual power RRP for the filtered input directions. **e** The optimized phase profiles of the RFG surfaces.

such that the corresponding permutation matrices $R_1$, $R_2$, and $R_3$ do not have overlapping entries, i.e., $\sum_{m,n} R_i[m,n] R_j[m,n] = 0$ if $i \neq j$. We chose the wavelengths $\lambda_1 = 0.70$ mm, $\lambda_2 = 0.75$ mm, and $\lambda_3 = 0.80$ mm to implement these refractive functions with an RFG comprising $K = 8$ surfaces. The refractive indices of the assumed RFG material at these wavelengths $(\lambda_1, \lambda_2, \lambda_3)$ are $n_1 = 1.6512$, $n_2 = 1.6518$, and $n_3 = 1.6524$,

respectively. The optimized thicknesses of the RFG surfaces are shown in Fig. 7b. As depicted in Fig. 7a (second row), the output angle error stays below $0.5°$ for all the input directions for the three unique refractive functions at the three wavelengths, demonstrating the success of wavelength multiplexing of refractive functions performed simultaneously through the same RFG. Note from the second row of

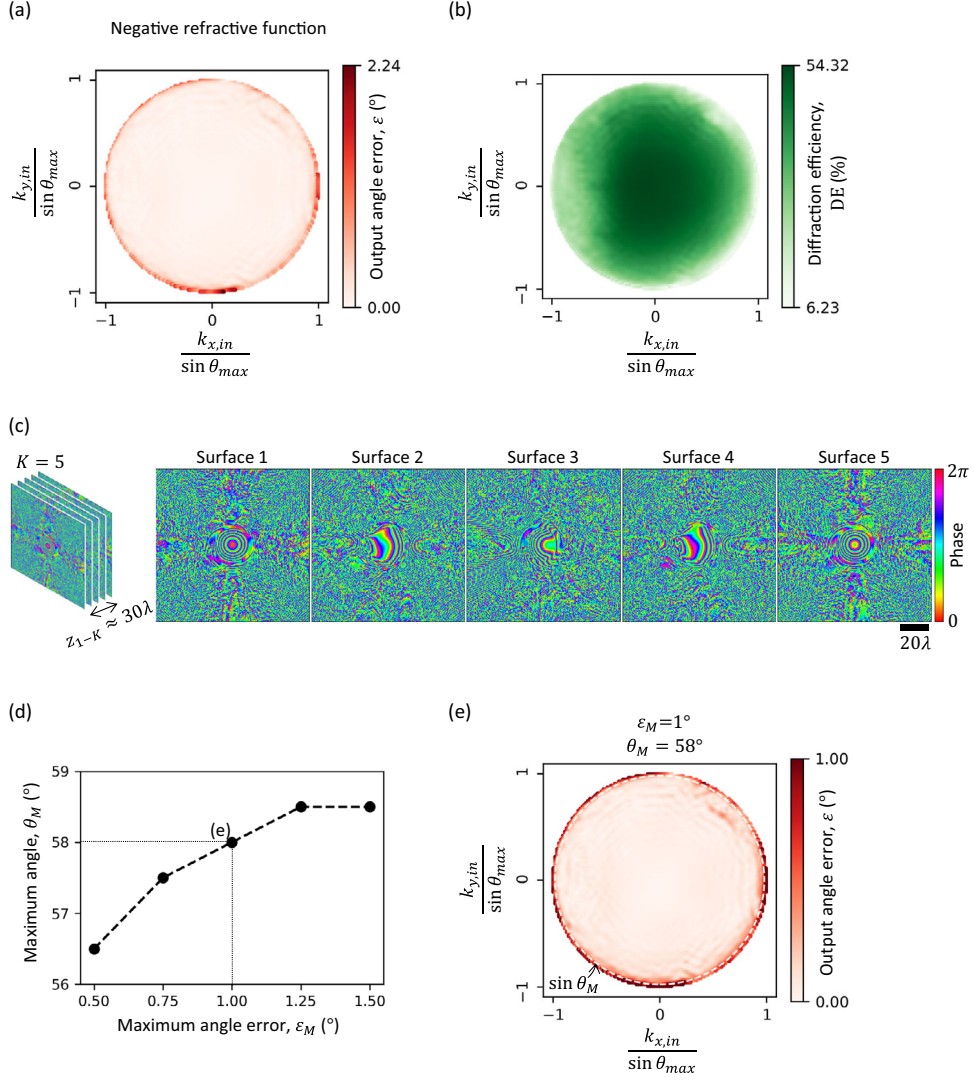

**Fig. 6 | Negative refractive function ($\theta_{target} = \theta_{in}$, $\varphi_{target} = \varphi_{in} + 180°$ for all $\theta_{in} < \theta_{max} = 60°$) using a $K = 5$ RFG design. a** Output angle error for all the input directions, sampled densely. **b** Diffraction efficiency for all the input directions. **c** The optimized phase profiles of the RFG surfaces. The distance $z_{ll}$ between consecutive surfaces is $\sim 6\lambda$, giving an axial distance of $z_{1-K} \approx 30\lambda$ between the first and the last transmissive surfaces. **d** The operating curve of the RFG design, showing $\theta_M$ (the maximum acceptable $\theta_{in}$) as a function of the maximum acceptable angle error, $\varepsilon_M$. **e** When $\varepsilon_M = 1°$, $\theta_M = 58°$, i.e., for all the input directions with $\theta_{in} < 58°$, the output angle error is less than $1°$.

Fig. 7a that the set of input directions for these refractive functions are not identical, since the grid-spacing depends on the wavelength (see Fig. 1b).

It is important to emphasize that this wavelength-multiplexed RFG design does not make use of the dispersion of the transmissive layers for its refractive function implementation accuracy; stated differently, even if we assume that the refractive indices of the assumed RFG material at these wavelengths ($\lambda_1$, $\lambda_2$, $\lambda_3$) are equal, i.e., $n_1 = n_2 = n_3 = n$, one could still perform wavelength-multiplexed refractive functions through an RFG design with the same level of accuracy and performance as shown earlier. Supplementary Fig. S2 compares the performance of an alternative design with flat dispersion, where $n = 1.6518$ was selected for all 3 wavelengths, revealing a statistically similar RFG performance as in Fig. 7. Supplementary Movie 2 also shows the far-field output intensity as the input wave direction is changed at these three wavelengths, together with the target patterns that follow the desired $f_i$ at the corresponding $\lambda_i$. These results indicate that the refractive function separation between different illumination wavelengths is based on the wavelength dependence of the free-space propagation kernel, and this unique capability does not need dispersion engineering of specialized materials, which is rather important for practical applications since one can readily work with almost any transmissive substrate that is available at a given desired spectral band.

## Experimental results

We experimentally demonstrated the success of programmable refractive function implementation at THz part of the spectrum with an illumination wavelength of $\lambda = 0.75$ mm. We designed an RFG comprising $K = 3$ phase-only surfaces to implement the negative refractive function for $\theta_{in} \leq \theta_{max} = 30°$. For this design, the width of the structured surfaces was selected as 80 mm, with a feature size of 0.4 mm, resulting in $\sim 0.12$ million independently optimizable phase features for the RFG design. The distance between neighboring surfaces was $\sim 16\lambda$, giving an axial span of $z_{1-K} \approx 32\lambda$ for the RFG design.

For resilience against potential misalignments during the experiment, the RFG design was "vaccinated" by applying random lateral shifts ($\Delta x$, $\Delta y$) to the surfaces during the digital training process.

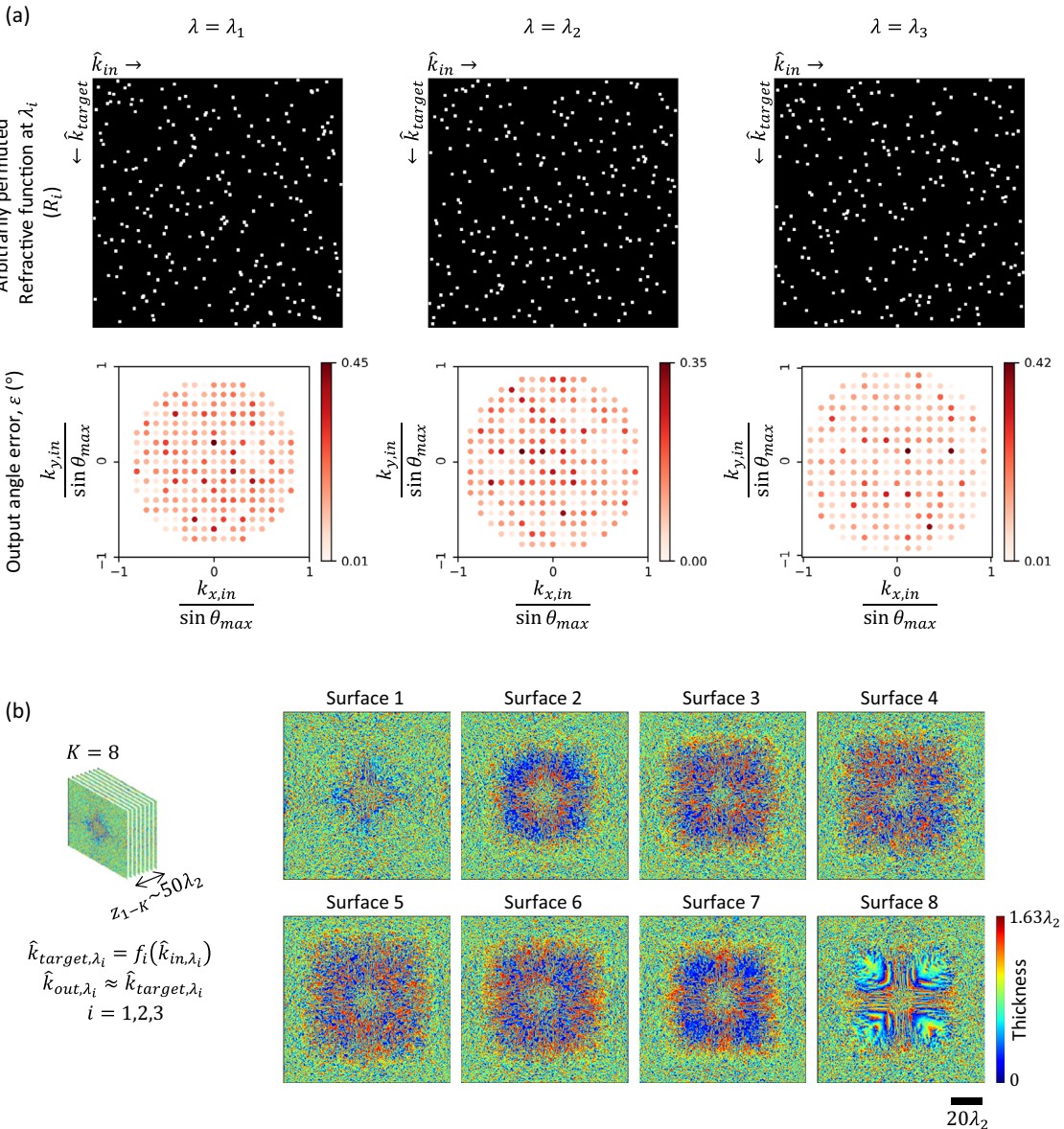

**Fig. 7 | Wavelength multiplexing of arbitrarily permuted refractive functions with an RFG. a** The matrices representing the targeted arbitrarily permuted refractive functions at three distinct wavelengths (top row). The bottom row shows, for a $K = 8$ RFG design, the error in the output angle as a function of the input direction at these three wavelengths. **b** The optimized thickness profiles of the RFG surfaces. The distance $z_{ll}$ between consecutive surfaces is $\sim 6\lambda_2$, giving an axial distance of $z_{1-K} \approx 50\lambda_2$ between the first and the last transmissive surfaces.

Similarly, the axial distances between the transmissive layers were also vaccinated against imperfections by adding random noise ($\Delta z$) in the optical forward model used during training. These random variables $\Delta x$, $\Delta y$, and $\Delta z$ were sampled from uniform distributions, i.e., Uniform($-0.15\lambda$, $0.15\lambda$). The optimized phase profiles of the resulting RFG surfaces are shown in Fig. 8a, together with the output angle errors and diffraction efficiencies obtained in numerical simulations.

After the deep learning-based supervised design of the desired RFG, the optimized surfaces were fabricated using a 3D printer and assembled, together with the input and output apertures, to form the physical RFG, as shown in Fig. 8b. This physically assembled RFG was experimentally tested with the system shown in Fig. 8c, which comprises a THz source and a THz scanning detector; see the "Methods" section for details.

Figure 9 shows the experimentally measured output intensities at an axial distance of $z = 80$ mm from the output aperture, together with

the corresponding simulation results for different input wave directions defined by $\theta_{in}$ and $\varphi_{in}$. During these experiments, the variation of $\theta_{in}$ was realized by moving the source horizontally along an arc (see Supplementary Fig. S1b), while the variation of $\varphi_{in}$ was implemented by in-plane rotation (relative to the source) of the RFG surfaces and the input and output apertures. To compensate for the relative rotation between the RFG and the detector plane, the fields of view (FOVs) corresponding to the experimental measurements were rotated by the same amount (in the opposite direction), as seen in Fig. 9. An additional calibration step to take into account the height of the source relative to the input aperture was also used; see Supplementary Fig. S1a.

Visual assessment of the output intensity patterns in Fig. 9 reveals a very good agreement between the simulated patterns and the measured output patterns. For quantitative analysis, we estimated the direction of the output waves ($\theta_{out}$, $\varphi_{out}$) from the first moments

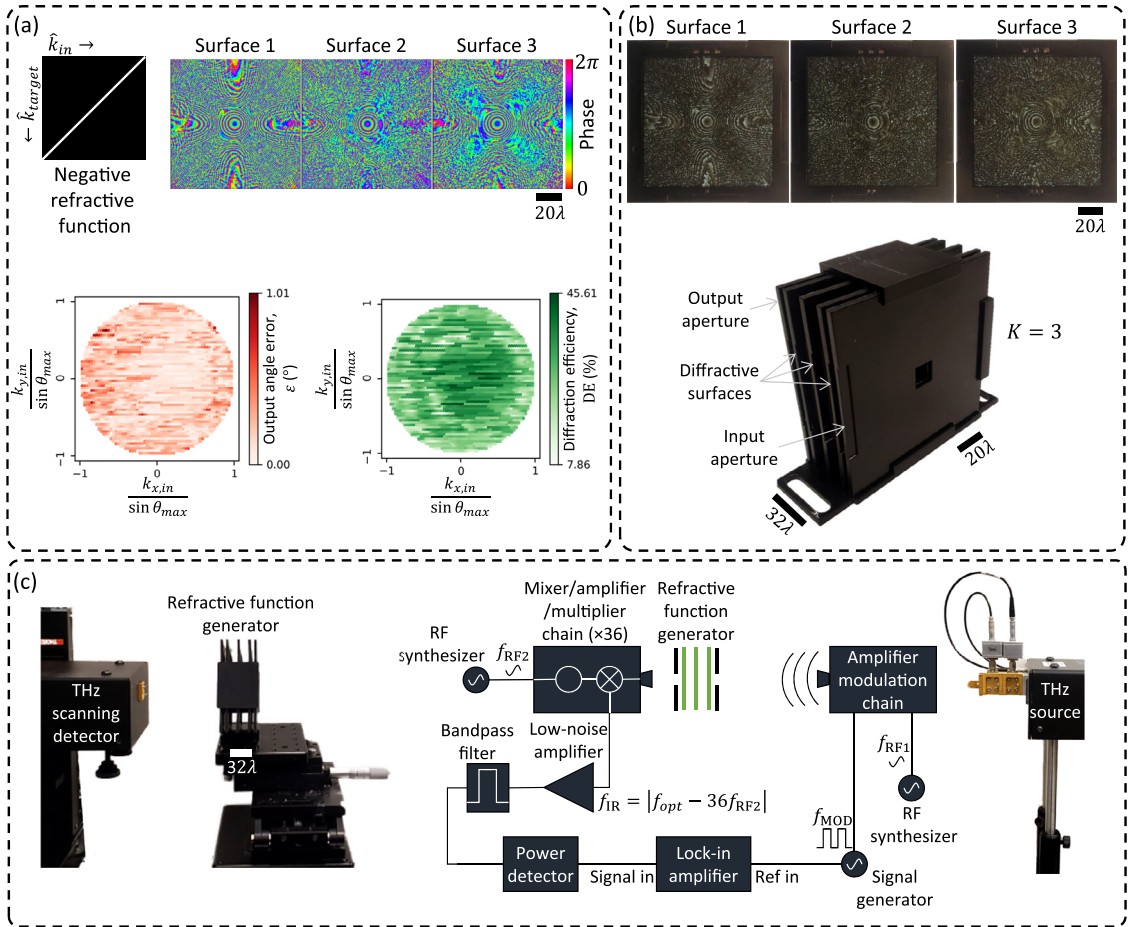

**Fig. 8 | Experimental demonstration of the negative refractive function at λ = 0.75 mm. a** The optimized phase profiles of a $K = 3$ RFG for implementing negative refractive function with $\theta_{max} = 30°$. Also shown are the output angle errors and the diffraction efficiencies obtained in simulation. **b** The RFG hardware, assembled from the structured surfaces and input/output apertures, fabricated using 3D-printing. **c** The THz setup comprising the source and the detector, together with the 3D-printed RFG.

(center-of-mass) of the diffracted output intensity patterns, which are marked by red dots in Fig. 9; also see Supplementary Fig. S1b. To quantify the mismatch between our simulations and experimental results, the angle between the output directions obtained from each simulation and the corresponding experiment ($\varepsilon_{sim-exp}$) is reported at the bottom of each panel corresponding to a $(\theta_{in}, \varphi_{in})$ combination. The minimum and maximum values of this angular error, $\varepsilon_{sim-exp}$, are $0.23°$ and $2.01°$, which occur at $(\theta_{in}, \varphi_{in}) = (19.81°, 276.48°)$ and $(\theta_{in}, \varphi_{in}) = (10.03°, 197.49°)$, respectively. The mean angular error is $1.04°$. These experimental results successfully demonstrate the proof of concept of our refractive function programming capability using the presented framework.

In addition to negative refractive function implementation, we also experimentally demonstrated a discrete permutation of input directions; see Fig. 10. We designed an RFG comprising $K = 3$ transmissive surfaces to implement a randomly selected permutation of five discrete input directions over a narrow angular range ($\theta_{max} = 4°$). For this design, the diffractive surfaces were 64 mm wide with a feature size of 0.4 mm, yielding 76,800 independently optimized phase elements. The axial separation between neighboring surfaces was $\sim 16\lambda$, resulting in a total device span of $z_{1-K} \approx 32\lambda$. The optimized phase profiles for the three diffractive surfaces are shown in Fig. 10a, along with the numerically evaluated angular errors and diffraction efficiencies corresponding to each input direction. The surfaces were fabricated via 3D printing and assembled into a complete RFG system, as shown in Fig. 10b. Figure 10c presents a comparison between the

simulated and measured intensity patterns at the detector plane, positioned 160 mm from the output aperture, for each of the input directions of interest. The green and red dots mark the center of the FOV and the first moment of the diffracted intensity pattern, respectively. The estimated angular mismatches $\varepsilon_{sim-exp}$ between our simulations and experimental results remain below 1.4° for all tested directions, confirming the successful implementation of the desired permutation of input angles by the fabricated RFG.

## Discussion
In this paper, we presented refractive function programming, i.e., on-demand engineering of the relationship between the input and output wave directions, by a collection of spatially optimized surfaces that comprises a physical RFG, axially spanning $\sim 15\lambda - 50\lambda$. We demonstrated the engineering of the output wave direction independently and arbitrarily for a set of input directions of interest, including arbitrarily filtered and permuted refractive functions as well as wavelength-multiplexed arbitrarily permuted refractive functions. We also reported other examples of refractive function programming, such as the negative refractive function, along with a proof-of-concept experimental demonstration of it at the THz part of the spectrum using a 3D-printed RFG. Related works on volumetric meta-optics have demonstrated the sorting of input light based on wavelength, polarization, and direction[55–57]; however, the dimensionality of the input direction space was limited to ≤5. More importantly, these approaches do not achieve arbitrary permutation

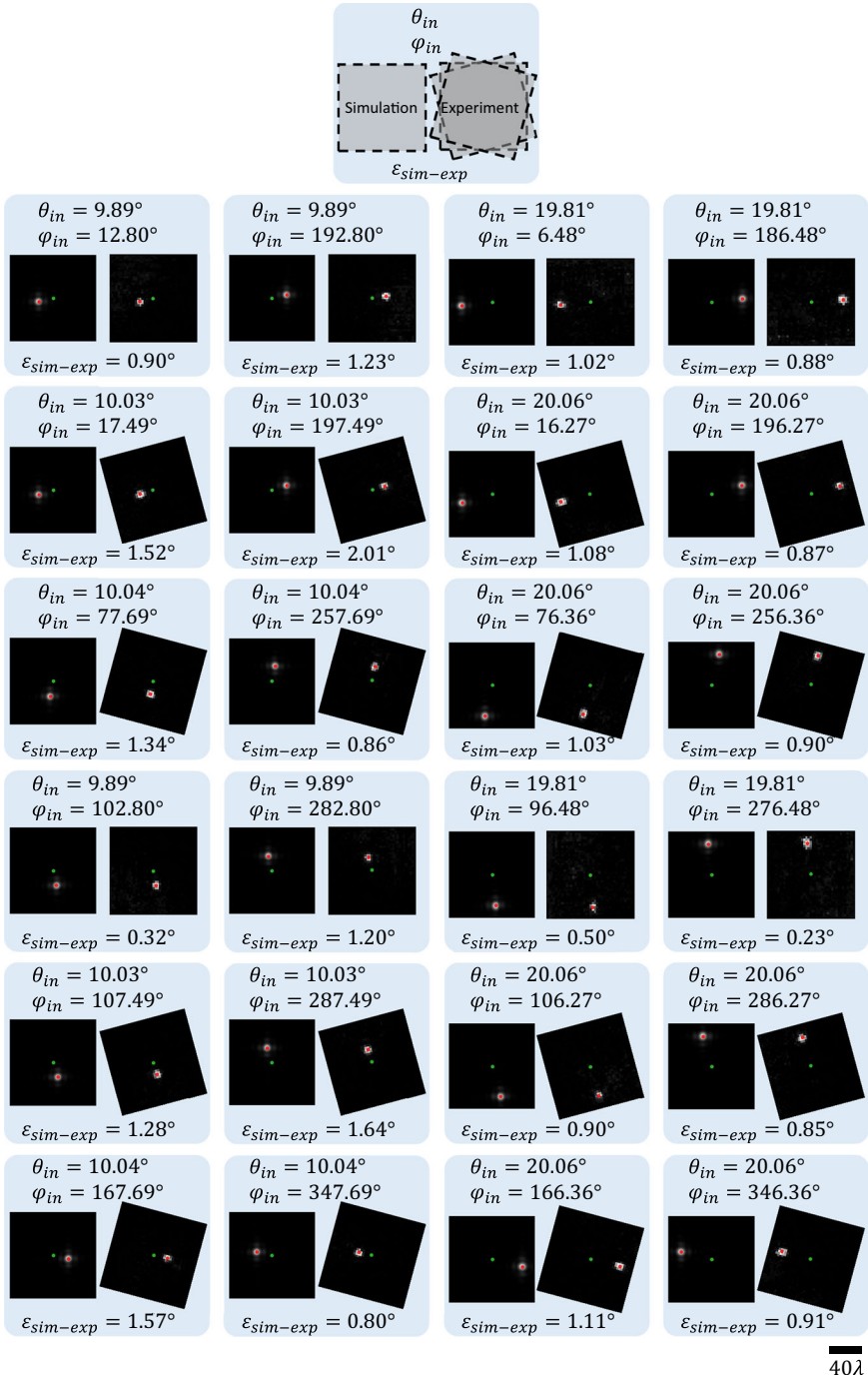

**Fig. 9 | Visualization and quantitative analysis of the experimental RFG results.** Each panel corresponds to the input direction defined by $(\theta_{in}, \varphi_{in})$ and compares the simulated and experimental diffraction patterns measured at a distance of $z = 80$ mm from the output aperture of the RFG (see Fig. 9c). The green dot marks the center $(0, 0)$ of the FOV and the red dot marks the first moment of the diffracted intensity pattern; also see Supplementary Fig. S1b. In each panel, the mismatch between the numerical simulation and the experimental result, defined as the angle $\varepsilon_{sim-exp}$ between the simulated output wave and the experimentally measured output wave, is also reported.

of input directions, i.e., they do not demonstrate arbitrary refractive function programming.

While our design approach draws on supervised learning tools, it does not aim to learn from data a general predictive model that maps an input direction $\hat{k}_{in}$ to a corresponding output direction $\hat{k}_{out}$. Instead, the desired functional mapping between input and output directions is known a priori and specified explicitly by the task—for example, an arbitrarily selected permutation or a negative refraction

function. The RFG is optimized to enforce this prescribed mapping between the directions of the input and output waves. The surface phase profiles are optimized to minimize the output angular error using deep learning tools such as error-backpropagation and gradient descent on a physics-based wave propagation model (see the "Methods" section for details). In the case of permutation tasks that are finite-dimensional, the input direction can only be selected from a fixed discrete grid, and in general, no generalization beyond this grid is

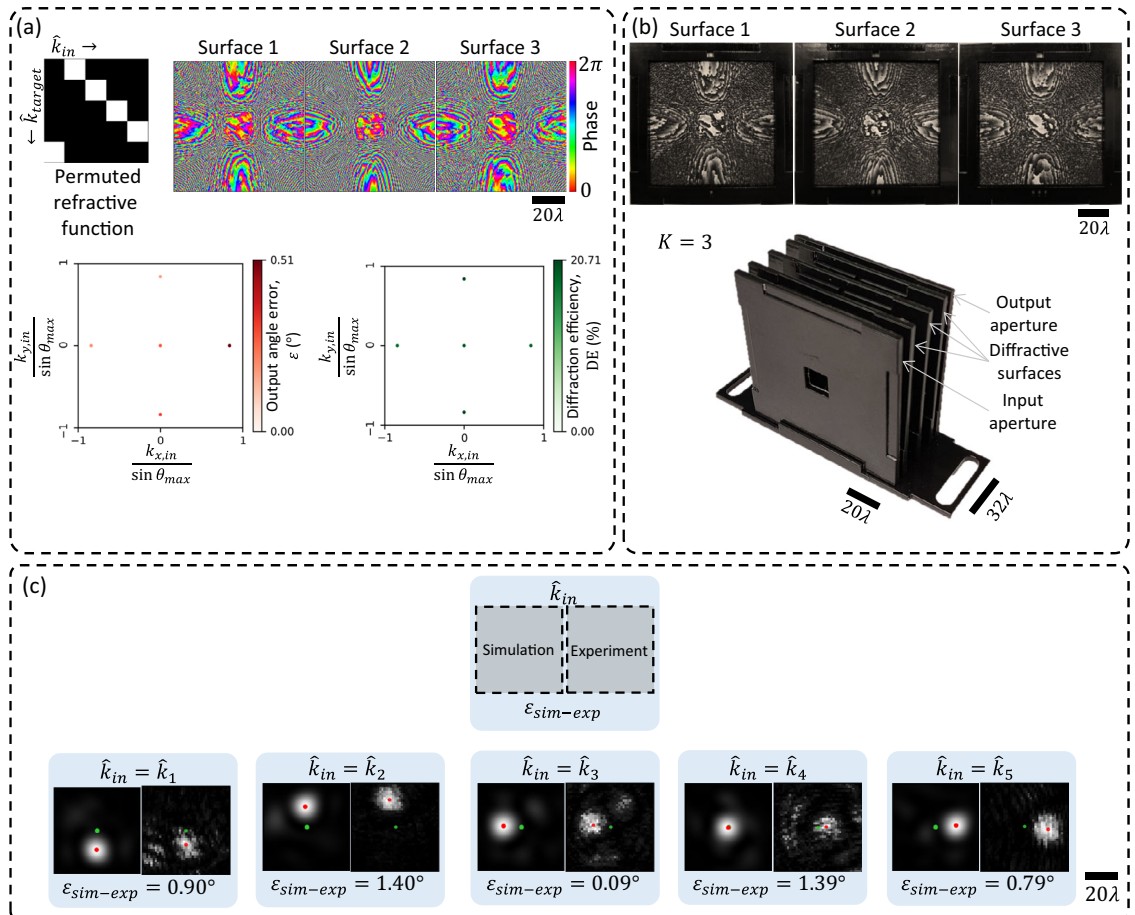

**Fig. 10 | Design, fabrication, and experimental validation of an RFG implementing a permutation refractive function. a** Top: a permutation refractive function (left) and the learned phase profiles of the three diffractive surfaces used to implement this permutation refractive function (right). Bottom: numerically evaluated output angle error and diffraction efficiency for each input direction. **b** Fabricated diffractive surfaces (top) and the 3D-printed assembly used for experimental testing (bottom). The assembly comprises the fabricated surfaces aligned between the input and output apertures. **c** Comparison between simulated and experimentally measured output intensity patterns 160 mm away from the output aperture for the five different input directions. The green dot marks the center (0, 0) of the FOV, and the red dot marks the first moment of the diffracted intensity pattern. The measured angular deviation $\varepsilon_{sim-exp}$ remains below 1.4° in all cases, demonstrating close agreement between the simulations and experimental results.

required. In contrast, for continuous mappings such as negative refractive function, the input direction can assume any value within a continuous domain. In this case, during the optimization/training, the input angles are randomly sampled from this continuous domain. Since the probability of the exact test directions being sampled during training is infinitesimal, it is valid to assume that the test directions differ from the training set—requiring generalization in the learning-theoretic sense. Thus, the problem of refractive function generation can require meaningful generalization depending on the nature of the function to be generated.

The supervised learning-based design approach we adopt offers several advantages. First, it naturally supports a 'vaccination' strategy, whereby the design is made resilient to anticipated deviations from ideal conditions (such as wavelength shifts or hardware misalignments) by introducing such deviations as random noise during training; see Fig. 3b. Second, our framework allows for a desired trade-off among competing performance metrics to be explicitly controlled through loss function engineering, as demonstrated in Fig. 4. To further highlight the versatility of our physics-based learning framework, we apply it to a classical task: design of light focusing elements. While an ideal lens defined by the phase profile $\psi(x,y) = \frac{2\pi}{\lambda}\sqrt{x^2 + y^2 + f^2}$ can provide ideal focusing at wavelength $\lambda$

and focal length $f$, its performance deteriorates when implemented using discretized phase elements. Under such constraints, our method can discover superior solutions. Supplementary Fig. S8 presents designs of focusing optics comprising $K$ diffractive surfaces, each consisting of phase elements discretized at $\lambda/2$ resolution and quantized into 8 uniformly spaced phase levels between 0 and $2\pi$, i.e., 3 phase bit-depth. For these simulations, we assume $\lambda = 400$ nm. The diameter of the diffractive surface(s) and the focal distance are assumed to be $288\lambda$ and $100\lambda$, respectively, resulting in a numerical aperture of 0.82. The diameter of the focal spot (region of interest, ROI) is set to be $\lambda$. To demonstrate the flexibility of tweaking the trade-off among performance metrics, we define two figures of merit relevant to energy localization: focusing efficiency (FE), which reports the ratio of optical power within the ROI to total input power, and power concentration ratio (PCR), which reports the ratio of power inside the ROI to that outside it. As shown in Supplementary Fig. S8, the learned diffractive focusing designs outperform the ideal lens phase profile (phase-wrapped and quantized) in both FE and PCR, even when using the same design degrees of freedom with $K = 1$. With $K = 2$ diffractive surfaces, the performance improves further, demonstrating the advantages of structural depth in our framework. Moreover, the ability to explicitly tune the trade-

off between FE and PCR via the loss function engineering further highlights the flexibility of our approach. This example illustrates how physics-based learning can push the boundaries of diffractive focusing beyond classical designs while remaining compatible with realistic fabrication constraints such as limited phase bit-depth.

While our RFG framework assumes phase-only modulation with unit transmission amplitude, this approximation is justified by the fact that absorption can be minimized through appropriate material selection. Since the same phase patterns can be rescaled to any operating wavelength by proportionally adjusting the physical dimensions of the diffractive design, such as the lateral pitch and axial separation, this design methodology remains broadly applicable across different spectral bands where low-loss materials are available. For example, high-resolution fabrication techniques in the visible regime allow for the use of low-loss dielectrics such as PMMA[58], whose negligible extinction coefficient enables the physical implementation of phase-only diffractive surfaces without significant absorption. To demonstrate this, we evaluated the permutation refractive function design of Fig. 4 (the one trained with $\eta = 30$) at a visible wavelength of 562.5 nm using the same optimized phase profiles, with all physical dimensions scaled according to the illumination wavelength. The diffractive surfaces were assumed to be fabricated from PMMA, with a refractive index of 1.4863 and an extinction coefficient of $2.27 \times 10^{-7}$ at 562.5 nm. As shown in Supplementary Fig. S9, the angular error and diffraction efficiency remained effectively unchanged from the original THz design, despite the use of eight transmissive layers. These results confirm that the design methodology is transferable across different spectral bands, and that absorption losses can be rendered negligible through careful selection of fabrication materials.

Power loss within an RFG can also arise from Fresnel reflections caused by refractive index discontinuities at the interfaces of the diffractive surfaces. For an RFG comprising 8 transmissive surfaces, there are 16 such optical interfaces, for example. Although our wave propagation simulations assume planar phase-only modulation and, therefore, do not explicitly model these reflections, their cumulative effect can be approximately estimated. For example, for PMMA in air ($n = 1.4863$), each interface reflects approximately $\left(\frac{n-1}{n+1}\right)^2 \approx 3.8\%$ of the incident power, resulting in an overall transmission factor of $(1 - 0.038)^{16} \approx 55\%$ due to interface reflections alone. In practice, these losses can be mitigated using anti-reflection coatings or index-matching layers, depending on the application requirements. Moreover, neglecting these reflections in the forward model introduces minimal error because the reflected waves are scattered by the elements of the preceding and proceeding structured layers since such secondary waves are considered noise and are not optimized for the intended output directions. This diffractive filtering of undesired secondary reflections is further supported by our THz experimental results (see Figs. 9 and 10), which show close agreement between the simulated and the measured angular output distributions.

Although the refractive functions that we discussed so far did not include many-to-one mappings between the input and output directions, i.e., more than one input directions do not result in the same output wave direction, this is not a restriction of the presented RFG framework. It is possible to design RFGs that implement 'many-to-one' refractive functions and also one-to-many mappings, giving rise to input-direction-specific programmed beam-splitting. To demonstrate this capability, we designed an RFG that performs a many-to-one transformation, in which all the input directions of interest are mapped to the same output direction. As shown in Supplementary Fig. S10, the optimized RFG performs this mapping with low output angle error across the full input range. Supplementary Movie 3 also shows the far-field output intensity as the input wave direction is swept, together with the corresponding target patterns that follow $f$.

As for the arbitrarily filtered and permuted refractive function reported in Fig. 5, the ratio of the filtered input directions was 90% (i.e.,

10% unfiltered input directions). We observed that as the ratio of the unfiltered directions within the desired refractive function increases, their output energy begins to spill into the filtered directions, causing the RRP for the latter to increase relatively, decreasing the refractive function approximation accuracy. Supplementary Fig. S3 shows the dependence of the average loss of various RFG designs as a function of the average ratio of the filtered directions; also see Supplementary Fig. S4, which reports an RFG design for an arbitrarily filtered and permuted refractive function with 80% of the input directions filtered. Part of the reason behind the relatively poor performance of RFGs designed for smaller ratios of filtered input directions might be due to the arbitrary selection of the filtered and unfiltered directions, causing the unfiltered directions to leak at the output aperture into their neighboring input directions that are desired to be filtered.

The RFG that was optimized for negative refractive function, shown in Fig. 6, was trained with a loss function that does not enforce any regularization related to diffraction efficiencies. As a result, the diffraction efficiencies were not uniform, with the high values only at lower angles of incidence. We present in Supplementary Fig. S11 an alternative RFG design that emphasizes improved diffraction efficiency across a wider range of input directions. The training loss function for this design was $L = L_{wf} + 5(1 - \theta_{in}/\theta_{max})(1 - DE)$, where $L_{wf}$ is the error between the output wavefront and the target wavefront and $DE$ is the diffraction efficiency (see Eqs. 11 and 13). For this new design, the diffraction efficiencies remain uniformly high over a wider range and fall to lower values only as $\theta_{in}$ approaches $\theta_{max}$. This improvement in diffraction efficiency is achieved at the cost of a small increase in the angular error, albeit near the edges of the input angular range only, i.e., $\theta_{in} \approx \theta_{max}$. A detailed analysis of the output diffraction efficiency performance is presented in Supplementary Fig. S11d, where we quantified the angular power distribution at the output aperture. At the bottom-right of Supplementary Fig. S11d, we report a "confusion matrix" that shows the diffraction efficiencies along all the output directions $\hat{k}_{out}$ for a given input direction $\hat{k}_{in}$. The dominantly flipped-diagonal structure of the confusion matrix (same as the structure of the matrix representing the negative refractive function, see Fig. 1) reveals the successful realization of the target refractive function. However, for input directions at the edge ($\theta_{in} \approx \theta_{max}$), there is significant leakage of power out of the target direction at the output aperture. The total output diffraction efficiency, obtained by summing over the rows of the "confusion matrix", can approach 99.24%, as shown in the accompanying plot; see top-right of Supplementary Fig. S11d.

In addition to the wavelength-multiplexed RFGs reported in our "Results" section, it is also possible to achieve polarization multiplexing of refractive functions using isotropic transmissive materials constituting the RFG; such an isotropic RFG design needs to be augmented with a separate, fixed polarizer array positioned between successive structured layers. In such an RFG design, different orthogonal polarization states at the input aperture can perform independent and arbitrarily selected refractive functions, and the use of a fixed/predetermined array of polarizers within the thin material volume would help us better utilize the different degrees of freedom of the RFG for different refractive functions. To demonstrate this capability, we designed an RFG comprising $K = 8$ transmissive surfaces, with a fixed array of orthogonal polarizers placed after the fourth surface; see Supplementary Fig. S12c. The structure was trained to implement two distinct permutation refractive functions at two orthogonal linear polarization states, i.e., horizontal ($p = 0°$) and vertical ($p = 90°$). Simulation results show that the output angle errors are limited to $\sim 0.5°$ across the entire input angular grid for both polarizations, confirming the feasibility of polarization-multiplexed refractive function generation using a shared diffractive volume. Supplementary Movie 4 also shows the far-field output intensity as the input wave direction is changed at these two polarization states,

together with the target patterns that follow the desired $f_i$ at the corresponding polarization.

It is also interesting to consider the directionality of refractive functions. For example, the RFG reported in Fig. 2, when used in the reverse axial direction, approximately performs the inverse of the forward refractive function $f$, albeit with a relatively large error since this reverse operation was not part of its training or design stage. The output angle errors of this RFG, together with the diffraction efficiencies, for propagation in the reverse axial direction are shown in Supplementary Fig. S5; here, the target directions in the reverse path are set by $f^{-1}$. With proper training that takes into account both the forward and backward operation of the RFG design, it should be possible to further optimize the RFG to perform a target refractive function ($f$) in the forward direction and its inverse ($f^{-1}$) in the reverse direction equally well. It should also be possible to design a unidirectional RFG which performs a desired refractive function in the forward direction only, while blocking all the directions of illumination in the reverse. While time-reversal symmetry prohibits unidirectional behavior in a lossless electromagnetic system without e.g., magneto-optical effects, this constraint does not strictly apply to RFGs. Importantly, the RFG is not a lossless system: power is inherently lost at the edges of the finite-sized diffractive surfaces due to diffraction and material absorption. This allows for asymmetry in how loss manifests across the forward and backward propagation paths. By leveraging this asymmetry into our learning framework, we can engineer the loss distribution differently for the two directions, achieving functionally unidirectional behavior *without* violating reciprocity. To better highlight this, we define a composite loss function $L = L_{wf}^{(f)} + \eta_f (1 - DE^{(f)}) + \eta_b DE^{(b)}$, where $L_{wf}$ defines the error between the output wavefront and the target wavefront (see Eq. 11), $DE$ denotes the diffraction efficiency (see Eq. 13). The superscripts ($f$) and ($b$) denote the forward and backward directions, respectively, while $\eta_f$ and $\eta_b$ are training hyperparameters defining the weights of the respective loss terms. This formulation allows for promoting high diffractive efficiency in the desired forward direction while simultaneously suppressing efficiency in the undesired reverse direction, i.e., creating an asymmetric diffractive system. Simulation results for two RFG designs trained using this loss function are shown in Supplementary Figs. S6 and S7. In the first case, we set $\eta_f = 0$ and $\eta_b = 1$, only penalizing the diffraction efficiency in the backward direction. This results in approximately six orders of magnitude difference between the median diffraction efficiencies in the forward and backward directions, in favor of the forward direction, illustrating the unidirectional behavior of our design. As desired, the angular error in the forward direction remains below 1°; see Supplementary Fig. 6. In the second case, reported in Supplementary Fig. 7, we simultaneously enforced high forward and low backward diffraction efficiencies by setting $\eta_f = 1$ and $\eta_b = 1$. This leads to further improvements in the forward diffraction efficiency (median 3.5%), while still maintaining strong suppression in the backward direction, with more than six orders of magnitude difference in median efficiencies between the forward and backward directions, once again showcasing the unidirectional behavior of our design. These results demonstrate that asymmetric loss engineering during the supervised learning phase enables functionally unidirectional RFGs, despite being implemented using passive and reciprocal materials.

In our framework, the shape of the input and output apertures—whether circular or square—is a configurable aspect of the RFG design. While the underlying physics may favor certain aperture shapes depending on the target refractive function, this choice is made prior to the training and incorporated into the physics model by defining the apertures accordingly. To illustrate this design flexibility, we present an additional design for the negative refractive function RFG reported in Fig. 6, where circular apertures of equal area are used instead of the square ones of the former design. As shown in Supplementary Fig. S13, this modification introduces a noticeable trade-off: although the

diffraction efficiency decreases, the angular error across the tested input directions improves significantly. Such performance trade-offs can be further tuned through appropriate loss function engineering, as discussed earlier (see Fig. 4).

We believe that the capability to program refractive functions within passive materials could unlock unprecedented opportunities in manipulating optical waves, with major implications for optical device and system design across various applications. For example, RFGs could support applications in multi-channel optical interconnects, where distinct input directions can be routed to designated output channels with high spatial precision, enabling compact and scalable free-space optical communication links. Additionally, the intrinsic spatial and directional mapping abilities of RFGs can be harnessed for secure optical encoding, embedding information in complex, task-specific refractive transformations that are difficult to intercept or replicate without complete knowledge of the RFG design. Beyond these applications, RFGs could also be used for wave routing and optical switching. By replacing bulky optical components with compact, task-adaptive surfaces, these programmable diffractive platforms can significantly reshape photonic system design, driving advances in, e.g., communication, sensing, and imaging technologies.

## Methods
### Model of wave propagation through an RFG
The RFG is assumed to comprise $K$ phase-only transmissive surfaces positioned axially at $z_1, z_2, \cdots, z_K$, respectively, between the input and the output apertures. The axial positions of the input aperture and the output aperture are denoted by $z_0$ and $z_{K+1}$, respectively. The physical wave propagation between the input and the output aperture is described by successive modulations of the wave by the transmissive surfaces, interleaved by free-space propagation between them. In the following notation, $w(x, y; z_l^-)$ denotes the wave incident on the diffractive surface at $z_l$, whereas $w(x, y; z_l^+)$ denotes the wave leaving the diffractive surface, after the corresponding phase modulation. The wave propagation through free-space between consecutive surfaces can be described by the Rayleigh–Sommerfeld diffraction integral. For $l = 1, 2, \cdots, K+1$

$$w(x, y; z_l^-) = \iint w(x', y'; z_{l-1}^+) h_{FSP}(x - x', y - y'; z_l - z_{l-1}) dx' dy' \quad (2)$$

Here $w(x, y; z_0^+) = w_{in}(x, y)$ is the input wave and $h_{FSP}(x, y; z)$ is the free-space propagation kernel for an axial distance of $z$:

$$h_{FSP}(x, y; z) = \frac{z}{r^2} \left( \frac{1}{2\pi r} + \frac{1}{j\lambda} \right) \exp \left( j \frac{2\pi r}{\lambda} \right) \quad (3)$$

where $r = \sqrt{x^2 + y^2 + z^2}$. At the structured surfaces, the incident waves are locally modulated by the corresponding transmittance values that are trainable. For $l = 1, 2, \cdots, K$

$$w(x, y; z_l^+) = t_l(x, y) w(x, y; z_l^-) \quad (4)$$

where $t_l(x, y)$ is the complex-valued transmittance function of the diffractive surface at $z_l$. One can write,

$$t_l(x, y) = a_l(x, y) \exp \left( j\psi_l(x, y) \right) \quad (5)$$

where $a_l(x, y)$ is the local amplitude/absorption and $\psi_l(x, y)$ is the phase-delay induced by the diffractive surface. For a material with negligible loss, $a_l(x, y) \approx 1$ whereas the phase delay is related to the local surface thickness $h_l(x, y)$ as follows:

$$\psi_l(x, y) = \frac{2\pi}{\lambda} (n - 1) h_l(x, y) \quad (6)$$

where $n$ is the refractive index of the material at the wavelength $\lambda$.

In our numerical simulations, free-space propagation of an optical field between successive transmissive surfaces was calculated using the angular spectrum method[59], which is a Fast Fourier transform-based implementation of the Rayleigh–Sommerfeld diffraction integral in Eq. (2). The fields/intensities were discretized using $\delta \approx 0.53\lambda$ along both $x$ and $y$, and sufficiently zero-padded to avoid aliasing[60].

## Design of an RFG

For a given pair of input and target directions $(\theta_{in}, \varphi_{in})$ and $(\theta_{target}, \varphi_{target})$ that satisfy the desired refractive function of interest, the wavefront incident on the input aperture at $z_0$ can be written as:

$$w(x, y; z_0^-) = \exp\left( j\frac{2\pi}{\lambda} \sin\theta_{in} (x \cos\varphi_{in} + y \sin\varphi_{in}) \right) \quad (7)$$

whereas

$$w_{in}(x, y) = w(x, y; z_0^+) = w(x, y; z_0^-) \operatorname{rect}\left(\frac{x}{D_a}\right) \operatorname{rect}\left(\frac{y}{D_a}\right) \quad (8)$$

Here $\operatorname{rect}(\frac{x}{D_a})\operatorname{rect}(\frac{y}{D_a})$ defines the input aperture and $D_a$ is its lateral width. Similarly, the wave leaving the output aperture can be written as:

$$w_{out}(x, y) = w(x, y; z_{K+1}^+) = w(x, y; z_{K+1}^-) \operatorname{rect}\left(\frac{x}{D_a}\right) \operatorname{rect}\left(\frac{y}{D_a}\right) \quad (9)$$

where $w(x, y; z_{K+1}^-)$ is calculated from $w(x, y; z_0^+)$ by successively applying Eqs. (2) and (4). The corresponding target wavefront can be written as:

$$w_{target}(x, y) = \exp\left( j\frac{2\pi}{\lambda} \sin\theta_{target} (x \cos\varphi_{target} + y \sin\varphi_{target}) \right) \\ \operatorname{rect}\left(\frac{x}{D_a}\right) \operatorname{rect}\left(\frac{y}{D_a}\right) \quad (10)$$

The transmissive surfaces were optimized by minimizing the error between the output wavefront and the target wavefront:

$$L_{wf} = 1 - \frac{\iint w_{out}(x, y) w_{target}^*(x, y) dx\, dy}{\left(\iint |w_{out}(x, y)|^2 dx\, dy\right)^{0.5} \left(\iint |w_{target}(x, y)|^2 dx\, dy\right)^{0.5}} \quad (11)$$

For tuning the output diffraction efficiencies of the RFGs for an arbitrarily permuted refractive function, the following batch loss function was used:

$$L_{batch} = \frac{1}{B} \sum_{i=1}^{B} L_{wf,i} + \eta \left\{ \left(1 - \frac{1}{B} \sum_{i=1}^{B} DE_i^\mu\right) + \left(1 - \frac{\min_i DE_i^\mu}{\max_i DE_i^\mu}\right) \right\} \quad (12)$$

where $B$ denotes the training batch size, the subscript $i$ indexes the examples of the input-output waves that satisfy the desired refractive function in a batch, and the diffraction efficiency (DE) along the target direction is defined as:

$$DE = \frac{\left| \mathcal{F}\{w_{out}\}(u_{target}, v_{target}) \right|^2}{\left| \mathcal{F}\{w_{in}\}(u_{in}, v_{in}) \right|^2} = \left| \frac{\iint w_{out}(x, y) w_{target}^*(x, y) dx\, dy}{\iint w_{in}(x, y) w_{in}^*(x, y) dx\, dy} \right|^2 \quad (13)$$

Here $u_{in} = \frac{k_{x,in}}{\lambda} = \frac{\sin\theta_{in} \cos\varphi_{in}}{\lambda}$, $v_{in} = \frac{k_{y,in}}{\lambda} = \frac{\sin\theta_{in} \sin\varphi_{in}}{\lambda}$ and similarly for $u_{target}$, $v_{target}$. $\mathcal{F}\{w_{out}\}(u, v)$ is the 2D Fourier transform of $w_{out}(x, y)$ evaluated at $(u, v)$. The first term within the curly brackets in Eq. (12) is intended to increase the output diffraction efficiency of the RFG, whereas the

second term is intended to reduce the nonuniformity in the diffraction efficiencies observed for different illumination directions. The exponent $\mu$ is a training hyperparameter, empirically set to 0.05. The hyperparameter $\eta$ can be tweaked to tune the resulting diffraction efficiencies, as shown in Fig. 4.

For training the arbitrarily filtered and permuted RFG of Fig. 5, the following loss function was used:

$$L = b(L_{wf} + \eta_1 \max(0, DE_{min} - DE) + \eta_2(1 - MP)) \\ + (1 - b)\eta_3 \max(0, RRP - RRP_{max}) \quad (14)$$

where $b = 1$ for the unfiltered illumination directions and $b = 0$ for the filtered ones. For the unfiltered directions, mode purity MP is defined as:

$$MP = \frac{\left| \mathcal{F}\{w_{out}\}(u_{target}, v_{target}) \right|^2}{\iint |\mathcal{F}\{w_{out}\}(u, v)|^2 du\, dv} = \frac{\left| \iint w_{out}(x, y) w_{target}^*(x, y) dx\, dy \right|^2}{\iint |w_{out}(x, y)|^2 dx\, dy} \quad (15)$$

For the filtered directions, the relative residual power RRP is defined as:

$$RRP = \frac{\iint |w_{out}(x, y)|^2 dx\, dy}{\iint |w_{in}(x, y)|^2 dx\, dy} \quad (16)$$

The minimum diffraction efficiency for the unfiltered directions $DE_{min}$ was set as 10%, whereas the maximum relative residual power for the filtered directions $RRP_{max}$ was set as 1%. The hyperparameters $\eta_1$, $\eta_2$ and $\eta_3$ determine the strength of different terms intended to enforce the diffraction efficiency to be above $DE_{min}$ while increasing the mode purity for the unfiltered directions as well as enforcing the relative residual power below $RRP_{max}$ for the filtered directions. We empirically set the values of these hyperparameters as $\eta_1 = \eta_2 = \eta_3 = 10$.

The optimizable features of an RFG design comprise the thickness variations $h_l(x, y)$ of the transmissive surfaces. The discretization interval for the RFG surfaces was $\delta \approx 0.53\lambda$, i.e., $h_l(m, n) = h_l(m\delta, n\delta)$. To limit the thickness variation to the maximum allowable value, $h_{max}$, they are obtained from the latent trainable parameters $\mu_l(m, n)$ as follows.

$$h_l(m, n) = (\mu_l(m, n) \bmod 1) \times h_{max} \quad (17)$$

We set $h_{max} = \frac{\lambda}{n-1}$ to allow for a phase modulation depth of $2\pi$. The latent variables $\mu_l(m, n)$ were initialized as zero at the beginning of the training.

For monochrome RFG designs reported in the manuscript, we assumed an operating wavelength of $\lambda = 0.75$ mm. For the negative refractive function generator, the input and output aperture width $D_a$ was 12 mm, whereas for the other examples, $D_a = 8$ mm. For the experimental results reported in Figs. 8 and 9, $\theta_{max} = 30°$. For the ones reported in Fig. 10, $\theta_{max} = 4°$, whereas $\theta_{max} = 60°$ was used for the rest of the results. The refractive index of the assumed RFG material (the same as the one used for experimental demonstration) was 1.6518 at 0.75 mm wavelength.

For the wavelength multiplexing examples, the assumed wavelengths were $\lambda_1 = 0.70$ mm, $\lambda_2 = 0.75$ mm and $\lambda_3 = 0.80$ mm, and the corresponding refractive indices were $n(\lambda_1) = 1.6512$, $n(\lambda_2) = 1.6518$ and $n(\lambda_3) = 1.6524$. To clarify, the physical dimensions were fixed across wavelengths to ensure consistent device geometry: the aperture width was set to $D_a = 8$ mm and each diffractive surface was 80 mm wide with $200 \times 200$ optimizable phase elements. In each corresponding figure, spatial dimensions are expressed in units of the

center wavelength ($\lambda_2 = 0.75$ mm) to facilitate comparison with other figures. For this wavelength multiplexing scheme, we set $h_{\max} = \max_i \frac{\lambda_i}{n(\lambda_i) - 1}$ to ensure that a full $2\pi$ phase modulation range is available at all the illumination wavelengths.

For the analysis of diffraction efficiency reported in Supplementary Fig. S11d, the grid of input and output directions is discretized at the diffraction-limited resolution $\lambda/D_a$ and subsequently flattened into a one-dimensional grid—a process we refer to as "unrolling"; see Supplementary Fig. S11.

The transmissive surfaces were optimized using the Adam optimizer[61] with a minibatch size of four and a learning rate of $10^{-3}$. We evaluated the mean loss of the trained model after the completion of each epoch and selected the trained model state at the end of the epoch corresponding to the lowest loss. The RFGs for negative refractive function were trained for 1000 epochs. The other RFG designs were trained for 6000 epochs, except for the ones reported in Supplementary Fig. S3, which were trained for 2000 epochs. The RFG models were implemented and trained using TensorFlow (version 2.4.1)[62] with Python 3.8. The training time depends on several factors, including the number of diffractive surfaces and the availability of GPU acceleration. For example, training a negative refractive function generator with $K = 5$ surfaces takes approximately 4 h on an NVIDIA RTX 4090 GPU. The fabrication of a single diffractive surface takes approximately 1.5 h on a Stratasys Objet30 V2 Pro 3D printer; the overall time can be significantly reduced by printing multiple surfaces simultaneously on the same build tray.

## Performance evaluation

The direction of the output wavefront $(\theta_{out}, \varphi_{out})$ was estimated by solving the following optimization problem to fit a uniform plane wave along $(\theta_{out}, \varphi_{out})$ to $w_{out}(x, y)$ using a gradient descent algorithm:

$$(\theta_{out}, \varphi_{out}) = \arg\max_{(\theta, \varphi)} \frac{\iint w_{out}(x,y) w_{\theta,\varphi}^*(x,y) dx\, dy}{\left(\iint |w_{out}(x,y)|^2 dx\, dy\right)^{0.5} \left(\iint |w_{\theta,\varphi}(x,y)|^2 dx\, dy\right)^{0.5}}$$

$$(18)$$

where

$$w_{\theta,\varphi}(x,y) = \exp\left( j \frac{2\pi}{\lambda} \sin\theta (x\cos\varphi + y\sin\varphi) \right) \mathrm{rect}\left(\frac{x}{D_A}\right) \mathrm{rect}\left(\frac{y}{D_A}\right)$$

$$(19)$$

The initial estimates for the optimization cycles, i.e., $\theta_{init}$ and $\varphi_{init}$, were derived from the Fourier transform of the output wavefront $\mathcal{F}\{w_{out}\}$ as follows:

$$(u_{init}, v_{init}) = \arg\max_{(u,v)} |\mathcal{F}\{w_{out}\}(u, v)|$$

$$(20)$$

$$\theta_{init} = \sin^{-1}\left( \lambda \sqrt{u_{init}^2 + v_{init}^2} \right)$$

$$(21)$$

$$\varphi_{init} = \tan^{-1} \frac{v_{init}}{u_{init}}$$

$$(22)$$

The angular error $\varepsilon$ between the output direction and the target direction was evaluated as follows:

$$\varepsilon = \cos^{-1}\left( \hat{k}_{out} \cdot \hat{k}_{target} \right)$$

$$(23)$$

where $\hat{k}_{out}$ and $\hat{k}_{target}$ are calculated from the corresponding $\theta$ and $\varphi$ values using Eq. (1), and $\hat{k}_{out} \cdot \hat{k}_{target}$ refers to their scalar product.

## Experimental setup

A modular amplifier (Virginia Diode Inc. WR9.0 M SGX) with multiplier chain (Virginia Diode Inc. WR4.3×2 WR2.2 × 2) and a compatible WR2.2 diagonal horn antenna from Virginia Diodes Inc. were used to generate continuous-wave radiation at 0.4 THz. This was accomplished by amplifying a 10 dBm RF input signal at $f_{RF1} = 11.1111$ GHz and multiplying it 36 times. To ensure low-noise data acquisition via lock-in detection, the AMC output was modulated at $f_{MOD} = 1$ kHz. The horn antenna's exit aperture was positioned at ~60 cm from the input aperture of the 3D-printed RFG so that the input THz wavefront was approximately planar. We used a Stratasys Objet30 V2 Pro printer for the 3D fabrication of the resulting RFG design. A single-pixel mixer from Virginia Diodes Inc. detected the diffracted THz radiation at ~80 mm away from the output aperture. The detected signal was down-converted to 1 GHz using a 10 dBm local oscillator signal at $f_{RF1} = 11.0833$ GHz fed into the mixer. The mixer, mounted on an X-Y positioning stage with two motorized linear stages (Thorlabs NRT100), scanned the output FOV using a $0.5 \times 0.25$ mm detector with 2 mm intervals. The down-converted signal was amplified by 40 dB using cascaded low-noise amplifiers (Mini-Circuits ZRL-1150-LN+), and unwanted noise was filtered out with a 1 GHz (+/−10 MHz) bandpass filter (KL Electronics 3C40-1000/T10-O/O). After measuring by a low-noise power detector (Mini-Circuits ZX47-60), the output voltage was then measured with a lock-in amplifier (Stanford Research SR830), using the $f_{MOD} = 1$ kHz modulation signal as a reference, and the amplifier readings were converted to linear scale. While estimating the output wave directions from the experimentally measured intensity patterns (see Supplementary Fig. S1b), only $5 \times 5$ pixels around the peak intensity was taken into account in calculating the first moment.

## Data availability

All the data needed to evaluate the conclusions of this work are present in the main text and the Supplementary Information. Additional data can be requested from the corresponding author (A.O.).

## Code availability

The deep learning models reported in this work used standard libraries and scripts that are publicly available in TensorFlow. A trained RFG model and the corresponding test code are available at: https://drive.google.com/drive/folders/1uAw-I0rVk8YKHPIWDZvUYcGz6HKOyV_D?usp=sharing.

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

## Acknowledgements

The Ozcan Research Group at UCLA acknowledges the support of the U.S. ARO (Army Research Office). The Jarrahi Group at UCLA acknowledges the support of the Institution of Engineering and Technology Harvey Engineering Research Prize.

## Author contributions

A.O. conceived and initiated the research. M.S.S.R. conducted the numerical simulations and analyses. M.S.S.R. and T.G. performed the experimental validation. M.S.S.R. and A.O. wrote the manuscript; M.S.S.R., T.G., M.J. and A.O. contributed to manuscript editing. A.O. supervised the research.

## Competing interests

The authors declare no competing interests.
