## [Transparent Peer Review file · Nature Communications]

Programming of refractive functions

Corresponding Author: Dr Aydogan Ozcan

Version 0:

Reviewer comments:

Reviewer #1

(Remarks to the Author)

This paper introduces arbitrary refractive function generators (RFGs) by cascading a series of diffractive surfaces, designed using supervised deep learning. A similar design was previously proposed by the same group for broadband THz pulse shaping. However, the functions demonstrated in this work are distinct and novel. Numerical simulations demonstrate fully arbitrary RFGs, filtered RFGs, negative RFGs, and wavelength multiplexing achieved via the proposed technique. Among these, the negative refractive function was further experimentally verified at THz wavelengths, with the device fabricated using 3D printing.

The functions achieved are of significant interest and importance in both optics and long-wavelength applications, such as millimeter-wave communications. The high accuracy and efficiency demonstrated in this study indicate strong potential for practical applications.

The paper is well-written, with a clear and rigorous presentation of theoretical design, numerical simulations, and experimental results, providing comprehensive and convincing evidence for the claimed conclusions. However, the methodology for deep learning-based design optimization is not entirely clear, as will be detailed below. With the following concerns properly addressed, I recommend the paper for publication in Nature Communications.

1. For readers unfamiliar with data-driven optimization, it is difficult to understand how the algorithm determines the required phase profiles of the diffractive surfaces from a given set of $[k_{in}, k_{out}]$. The primary explanation for this is given in the first sentence of paragraph 3 on page 3: "The design of an RFG" Supervised deep learning generally implies training a general network with labeled data, such that once trained, it can immediately generate solutions for any newly provided set of $[k_{in}, k_{out}]$. If my understanding is correct, the 'deep learning' in this work refers to an optimization process tailored for a specific set of $[k_{in}, k_{out}]$, rather than a general predictive model. I suggest providing a clearer explanation of the algorithm's working principles, especially for readers unfamiliar with the Adam optimizer.

2. It should be explicitly stated that the results in the sub-sections before "Experimental Results" are obtained through numerical simulations rather than experimental measurements. This distinction should also be emphasized in the supplementary videos, as they closely resemble experimental data rather than theoretical predictions.

3. The widths of the diffractive surfaces are only mentioned in the experimental section and are not discussed in the theoretical calculations. It is unclear how they relate to the aperture size and how they impact performance. Since the size also determines the number of phase elements and thus affects the scale of unknown parameters, a more detailed explanation is warranted.

4. Rectangular apertures were used for both the input and output in this study. However, optical beams generally have a circular cross-section, so using circular apertures could improve efficiency. Could the authors comment on the potential differences or the rationale behind choosing rectangular apertures?

5. While neglecting amplitude modulation is a reasonable approximation in this design, the number of diffractive surfaces used may significantly impact transmission loss and diffraction efficiency. It would be beneficial to consider this effect in the analysis.

6. For the multiplexing function, three wavelengths are involved. The device has numerous wavelength-dependent

parameters, such as the discretization interval of the diffractive surfaces and the distances between them. How were these parameters defined and optimized in the multiplexing design?

7. In Fig. 10, why are the experimental figures in the first and fourth rows not rotated? According to the experimental protocol, they should be rotated if ϕ_{in} is not an integer multiple of 90° .

8. In the discussion section, the authors mention the possibility of designing a unidirectional RFG that allows only forward transmission while blocking backward propagation. However, such functionality appears unfeasible, as it requires breaking time-reversal symmetry, which is typically achieved via magneto-optical effects. Diffractive surfaces, in general, do not violate time-reversal symmetry.

9. Finally, I suggest adding a brief background discussion on the potential applications of programmable diffractive functions in various fields.

Reviewer #2

(Remarks to the Author)

The noteworthy results of this paper are:

1. the specific functionality presented: a negative refractive index volume, and wavelength-based electromagnetic multiplexing - the general claim being the implementation of the first "refractive function generator"
2. the engineering and design approach of the authors to implement a volume, passive, linear metamaterial in the THz regime using "supervised deep learning"

The major categories where this work seems to fit best are in electromagnetic inverse design and linear metamaterials.

The work is presented reasonably well with few clerical errors, and the breadth of the optimization of inputs is quite large. The mathematical framework underpinning the optimization is well elucidated, and the details of the optimization hyperparameters are provided. The experiment seems to agree relatively well with simulation qualitatively.

However, this work to me does not seem to produce a significant scientific improvement in either of the fields. The design is done using a well understood electromagnetic model (Fourier optics), using a gradient-based optimization method that the Ozcan group and others have previously presented. While the figure of merit seems to be somewhat different, the actual physics being simulated do not seem to be improved or changed at all. In addition, the performance of the devices themselves is relatively low, and the applicability of the devices to other adjacent fields also seems low.

Previous publications as an example (not exhaustive) by the Ozcan group include:

1. "all-optical neural networks..." DOI: 10.1126/science.aat8084
2. "computational imaging without a computer..." DOI: 10.1186/s43593-022-00012-4
3. "cascadeable all-optical NAND gates..." DOI: 10.1038/s41598-022-11331-4

From a performance perspective, the maximum efficiency in the case of the negative refractive index material only reaches < 60% in simulation with low incident angle illumination. While there are many angles tested, this is still a discrete optimization, and there is significant symmetry in the negative refraction optical function. It seems there is some functional form in this case that the optimization is approaching imperfectly due to the non-convex nature of electromagnetic inverse design.

In the more general and challenging case of an 'arbitrary' angle mapping, the efficiency falls quite far to < 5%. Contrary to the authors' insinuation that a 4-degree error with a 1% change in wavelength is good, it actually seems to be quite normal in the realm of linear metamaterials, but this is to be expected as the authors did not attempt to correct for chromatic aberration.

In some sense, the relatively low performance of the devices could be excused to some degree if the functionality seemed to demonstrate some application or use in the real world. The authors claim that "many-to-one" or "one-to-many" functionalities could be easily achieved without any demonstration. If it was so easily achieved, why is there no demonstration of this application? This would be great if true as it is definitely high impact and of fundamental interest to many beam combiners in AR/VR applications. These functions in general are thought to be quite difficult to achieve due to fundamental limitations of etendue, which apply to linear, passive systems.

Other groups, notably the Faraon group have demonstrated similar functionalities that could be considered "refractive function generators" using inverse design in the mid-infrared range and simulated them in the visible range - notably focusing on functionalities which are relevant to the adjacent fields such as color routing for image sensors or wavefront sensing eg:

1. "Multifunctional volumetric meta-optics..." DOI: 10.1364/OPTICA.384228
2. "Multi-dimensional wavefront sensing..." DOI: 10.1364/OE.492440

The authors may also wish to consult a previous work which leverages this technique in an extremely low contrast system in the optical regime

1. "Aperiodic Volume Optics" DOI: 10.1038/nphoton.2009.290

Finally, while there is significant detail in terms of the mathematics and hyperparameters, there does not seem to be any detail regarding the architecture of the tensorflow network used to implement the surfaces. It is customary in papers which utilize neural networks to disclose the architecture used in terms of layers used, activation functions, skips, pooling, etc. Most neural network papers have a figure which shows the architecture used in addition to a description.

Reviewer #3

(Remarks to the Author)

This paper introduces a compact refractive function generator that enables arbitrary light refraction by permuting the mapping between input and output directions, including multiplexed control via different wavelengths. The design is implemented using cascaded transmissive layers with optimized phase profiles and is experimentally demonstrated at terahertz frequencies. The work is innovative and opens new possibilities in static, function-specific optical control. While the work presents an innovative approach with promising potential in static, function-specific optical control, some expressions in the paper raise questions. My comments and questions are listed as follows

1. The term "programmable" is widely used in the research field of metasurface and integrated photonics; it typically implies dynamic, reconfigurable control at the level of individual elements. However, the proposed device has a fixed function once it is fabricated, which might not align with the usual interpretation of "programmable." The authors are advised to clarify this terminology and consider a more accurate descriptor that better reflects the nature of the device.
2. While wavelength-selective behavior is claimed in the paper, the device does not appear to function as a filter. Instead of absorbing or reflecting energy, the device redistributes the incident wave directionally. If the disappearance of a signal at a particular wavelength is due to spatial diffusion rather than reflection or absorption, the term "refractive function manipulation" may not be entirely appropriate. In other words, we cannot use the equivalent refractive index to calculate the energy distribution ratios between absorption, refraction, and reflection. The authors should clarify the physical mechanism behind this behavior to avoid any confusion.
3. Based on the above two reasons, I believe that while the work is innovative, the title "Programmable refractive functions" is misleading. I suggest revising the title to better reflect the actual functionality.
4. Since the authors aim to emphasize the editability of arbitrary output directions, it would be helpful to specify the time required to design and fabricate a device for a given refractive function. This information is crucial as it directly impacts the scalability and usability of the device in real-world scenarios.
5. The experimental results show relatively low diffraction efficiency, which is worth further analysis. The authors should investigate the underlying causes of this issue, such as phase quantization, fabrication errors, or design limitations. A discussion of possible improvements would enhance the clarity and completeness of the manuscript.
6. To strengthen the manuscript, it would be valuable to design a classical diffractive optical element, such as a focusing lens, using the proposed method. A comparison of its performance with traditional diffractive optical elements in terms of focusing efficiency, aberration, and compactness would provide useful context and highlight the advantages of the proposed approach.

Version 1:

Reviewer comments:

Reviewer #1

(Remarks to the Author)

The authors have thoroughly addressed my questions. The newly added data and images are clear and sufficient to support the conclusions. The paper is recommended to be accepted without further comments.

(Remarks on code availability)

Reviewer #2

(Remarks to the Author)

Thank you to the authors for providing great responses to the questions! No further comments, I believe the manuscript is in good shape!

(Remarks on code availability)

The code appears to implement all of the major functions described in the paper. I have not run it but provided readme seems to be enough to get it started.

Reviewer #3

(Remarks to the Author)

In the revised manuscript, the authors clarified the use of “programmable” by distinguishing between one-time and dynamic programmability, and revised the title accordingly. They also provided new simulation results on diffraction efficiency and designed a classical focusing element for comparison, which effectively demonstrates the advantages of their method. The physical mechanism behind wavelength-dependent behaviour was also clearly explained.

I think the authors have well addressed my concerns with concrete improvements. I strongly recommend acceptance for publication.

(Remarks on code availability)

We sincerely thank the referees for their reviews and the constructive feedback that we have received on our manuscript “**Programmable refractive functions**” (revised title “**Programming of Refractive Functions**”) submitted to *Nature Communications* (NCOMMS-25-04448).

As detailed below, we have revised our manuscript in response to the reviewers’ comments. The original referee comments are shown in black color, whereas for ease of communication, our answers are provided in blue. The changes that we have made in the manuscript text are highlighted in yellow.

Summary of our Revisions:

We have revised our manuscript according to the reviewers’ comments and **added 9 new figures and 2 new videos**, which will be detailed in our specific responses listed below.

- Following the recommendation of a referee, we have revised the title of our manuscript to “**Programming of Refractive Functions**”
- We have added **new results on the design of focusing optics using our approach** to highlight its advantages compared to traditional designs.
- To address the reviewer comments, we have added new results and discussions on the following:
 - **Design of many-to-one refractive function generators (RFGs)**
 - **Design of unidirectional RFGs**
 - **Design of polarization multiplexed permutation RFGs**
 - **Analysis of diffraction efficiency of RFGs**
- We have added **new experimental results on permuted refractive function generation**

As a quick summary, the following items have been revised and/or added, all highlighted in yellow in our manuscript files:

Revised sub-sections:

- **Changes to the main text:**
 - Title
 - Abstract
 - Introduction
 - Results
 - Discussion
 - Methods

Renumbered Figures:

Previous	New
Fig. 5	Fig. 4
Fig. 6	Fig. 5
Fig. 7	Fig. 6
Fig. 8	Fig. 7
Fig. 9	Fig. 8
Fig. 10	Fig. 9

Revised Figures (the figure # refers to the number after renumbering):

- **Fig. 3:** Wavelength sensitivity and compactness of RFGs.

New Figures Added:

- **Fig. 10:** Design, fabrication, and experimental validation of an RFG implementing a permuted refractive function.
- **Supplementary Fig. S6:** Unidirectional RFG design for arbitrary permuted refractive function by only suppressing backward transmission.
- **Supplementary Fig. S7:** Unidirectional RFG design for arbitrary permuted refractive function by both boosting forward transmission and suppressing backward transmission.
- **Supplementary Fig. S8:** Diffractive focusing optics using physics-based learning of cascaded surfaces.
- **Supplementary Fig. S9:** Performance of an RFG design implemented in the visible regime using a low-loss material.
- **Supplementary Fig. S10:** Many-to-one refractive function realization using an RFG.
- **Supplementary Fig. S11:** Efficiency-enhanced RFG design for negative refractive function. (a) Output angle error for all the input directions, sampled densely within θ_{max} .
- **Supplementary Fig. S12:** Polarization multiplexing of arbitrarily permuted refractive functions with an RFG by using a preset array of polarizers.
- **Supplementary Fig. S13:** Negative refractive function RFG with circular aperture. (a) Output angle error for all the input directions, sampled densely within θ_{max} .

New Supplementary Videos Added:

- **Supplementary Video 3:** Far-field intensity patterns of a many-to-one refractive function generator for different input directions.
- **Supplementary Video 4:** Far-field intensity patterns corresponding to polarization multiplexing of arbitrarily permuted refractive functions.

Reviewer #1:

This paper introduces arbitrary refractive function generators (RFGs) by cascading a series of diffractive surfaces, designed using supervised deep learning. A similar design was previously proposed by the same group for broadband THz pulse shaping. However, the functions demonstrated in this work are distinct and novel. Numerical simulations demonstrate fully arbitrary RFGs, filtered RFGs, negative RFGs, and wavelength multiplexing achieved via the proposed technique. Among these, the negative refractive function was further experimentally verified at THz wavelengths, with the device fabricated using 3D printing.

The functions achieved are of significant interest and importance in both optics and long-wavelength applications, such as millimeter-wave communications. The high accuracy and efficiency demonstrated in this study indicate strong potential for practical applications.

The paper is well-written, with a clear and rigorous presentation of theoretical design, numerical

simulations, and experimental results, providing comprehensive and convincing evidence for the claimed conclusions. However, the methodology for deep learning-based design optimization is not entirely clear, as will be detailed below. With the following concerns properly addressed, I recommend the paper for publication in Nature Communications.

- We sincerely thank the reviewer for their positive and constructive evaluation.

1. For readers unfamiliar with data-driven optimization, it is difficult to understand how the algorithm determines the required phase profiles of the diffractive surfaces from a given set of $[\mathbf{k}_{in}, \mathbf{k}_{out}]$. The primary explanation for this is given in the first sentence of paragraph 3 on page 3: “The design of an RFG ...” Supervised deep learning generally implies training a general network with labeled data, such that once trained, it can immediately generate solutions for any newly provided set of $[\mathbf{k}_{in}, \mathbf{k}_{out}]$. If my understanding is correct, the ‘deep learning’ in this work refers to an optimization process tailored for a specific set of $[\mathbf{k}_{in}, \mathbf{k}_{out}]$, rather than a general predictive model. I suggest providing a clearer explanation of the algorithm’s working principles, especially for readers unfamiliar with the Adam optimizer.

- We sincerely thank the reviewer for the suggestion. We have added the following paragraph to the Discussion section to clarify the differences between our approach and predictive modeling:

“... While our design approach draws on supervised learning tools, it does not aim to learn from data a general predictive model that maps an input direction $\hat{\mathbf{k}}_{in}$ to a corresponding output direction $\hat{\mathbf{k}}_{out}$. Instead, the desired functional mapping between input and output directions is known a priori and specified explicitly by the task - for example, an arbitrarily selected permutation or a negative refraction function. The RFG is optimized to enforce this prescribed mapping between the directions of the input and output waves. The surface phase profiles are optimized to minimize the output angular error using deep learning tools such as error-backpropagation and gradient descent on a physics-based wave propagation model (see the Methods section for details). In the case of permutation tasks that are finite-dimensional, the input direction can only be selected from a fixed discrete grid, and in general no generalization beyond this grid is required. In contrast, for continuous mappings such as negative refractive function, the input direction can assume any value within a continuous domain. In this case, during the optimization/training, the input angles are randomly sampled from this continuous domain. Since the probability of the exact test directions being sampled during training is infinitesimal, it is valid to assume that the test directions differ from the training set—requiring generalization in the learning-theoretic sense. Thus, the problem of refractive function generation can require meaningful generalization depending on the nature of the function to be generated.”

2. It should be explicitly stated that the results in the sub-sections before “Experimental Results” are obtained through numerical simulations rather than experimental measurements. This distinction should also be emphasized in the supplementary videos, as they closely resemble experimental data rather than theoretical predictions.

- Following the reviewer’s suggestion, we have added the following clarification to the Results section:

“The results and analyses presented in the subsections leading up to the ‘Experimental results’ as well as the animations presented in the Supplementary Videos are based on numerical simulations. ...”

3. The widths of the diffractive surfaces are only mentioned in the experimental section and are not discussed in the theoretical calculations. It is unclear how they relate to the aperture size and how they impact performance. Since the size also determines the number of phase elements and thus affects the scale of unknown parameters, a more detailed explanation is warranted.

- Following the reviewer's suggestion, we have added the following clarification to the Results section:

"...Unless otherwise stated, each diffractive surface in these simulations consists of 200×200 independently optimized phase features/elements. Each phase element spans an area of $0.53\lambda \times 0.53\lambda$, resulting in a total surface width of 106λ . For arbitrary permutation refractive functions, both the input and output apertures have a width of 10.6λ ; for negative refractive functions, this width is set to 15.9λ . The separation between consecutive planes—whether they contain input/output apertures or diffractive surfaces—is set to 6λ , unless specified otherwise. The maximum input polar angle θ_{max} is assumed to be 60° ."

4. Rectangular apertures were used for both the input and output in this study. However, optical beams generally have a circular cross-section, so using circular apertures could improve efficiency. Could the authors comment on the potential differences or the rationale behind choosing rectangular apertures?

- We thank the reviewer for this opportunity to clarify our results. We have added the following paragraph in the Discussion section, supported by **the newly added supplementary figure Fig. S13**, to elaborate on this choice:

*"...In our framework, the shape of the input and output apertures - whether circular or square - is a configurable aspect of the RFG design. While the underlying physics may favor certain aperture shapes depending on the target refractive function, this choice is made prior to the training and incorporated into the physics model by defining the apertures accordingly. **To illustrate this design flexibility, we present an additional design for the negative refractive function RFG reported in Fig. 6, where circular apertures of equal area are used instead of the square ones of the former design. As shown in Supplementary Fig. S13, this modification introduces a noticeable trade-off: although the diffraction efficiency decreases, the angular error across the tested input directions improves significantly.** Such performance trade-offs can be further tuned through appropriate loss function engineering, as discussed earlier (see Fig. 4)."*

5. While neglecting amplitude modulation is a reasonable approximation in this design, the number of diffractive surfaces used may significantly impact transmission loss and diffraction efficiency. It would be beneficial to consider this effect in the analysis.

- We appreciate the reviewer's valuable suggestions. We have added **a new supplementary figure Fig. S9**, together with the following Discussion section paragraphs to address this concern:

"...While our RFG framework assumes phase-only modulation with unit transmission amplitude, this approximation is justified by the fact that absorption can be minimized through appropriate material selection. Since the same phase patterns can be rescaled to any operating wavelength by proportionally adjusting the physical dimensions of the diffractive design, such as the lateral pitch and axial separation, this design methodology remains broadly applicable across different spectral bands where low-loss materials are available. For example, high-resolution fabrication techniques in the visible regime allow for the use of low-loss dielectrics such as PMMA⁵⁸, whose negligible extinction coefficient enables the physical implementation of phase-only diffractive surfaces without significant

absorption. To demonstrate this, we evaluated the permutation refractive function design of Fig. 4 (the one trained with $\eta = 30$) at a visible wavelength of 562.5 nm using the same optimized phase profiles, with all physical dimensions scaled according to the illumination wavelength. The diffractive surfaces were assumed to be fabricated from PMMA, with a refractive index of 1.4863 and an extinction coefficient of 2.27×10^{-7} at 562.5 nm. As shown in **Supplementary Fig. S9**, the angular error and diffraction efficiency remained effectively unchanged from the original THz design, despite the use of 8 transmissive layers. These results confirm that the design methodology is transferable across different spectral bands, and that absorption losses can be rendered negligible through careful selection of fabrication materials.

Power loss within an RFG can also arise from Fresnel reflections caused by refractive index discontinuities at the interfaces of the diffractive surfaces. For an RFG comprising 8 transmissive surfaces, there are 16 such optical interfaces, for example. Although our wave propagation simulations assume planar phase-only modulation and, therefore, do not explicitly model these reflections, their cumulative effect can be approximately estimated. For example, for PMMA in air ($n = 1.4863$), each interface reflects approximately $\left(\frac{n-1}{n+1}\right)^2 \approx 3.8\%$ of the incident power, resulting in an overall transmission factor of $(1 - 0.038)^{16} \approx 55\%$ due to interface reflections alone. In practice, these losses can be mitigated using anti-reflection coatings or index-matching layers, depending on the application requirements. Moreover, neglecting these reflections in the forward model introduces minimal error because the reflected waves are scattered by the elements of the preceding and proceeding structured layers since such secondary waves are considered noise and are not optimized for the intended output directions. **This diffractive filtering of undesired secondary reflections is further supported by our THz experimental results (see Figs. 9 and 10), which show close agreement between the simulated and the measured angular output distributions.**

6. For the multiplexing function, three wavelengths are involved. The device has numerous wavelength-dependent parameters, such as the discretization interval of the diffractive surfaces and the distances between them. How were these parameters defined and optimized in the multiplexing design?

- We have added the following details to the Methods section to better explain the parameterization with respect to wavelength for the wavelength-multiplexed refractive functions:

“...For the wavelength multiplexing examples, the assumed wavelengths were $\lambda_1 = 0.70$ mm, $\lambda_2 = 0.75$ mm and $\lambda_3 = 0.80$ mm, and the corresponding refractive indices were $n(\lambda_1) = 1.6512$, $n(\lambda_2) = 1.6518$ and $n(\lambda_3) = 1.6524$. To clarify, the physical dimensions were fixed across wavelengths to ensure consistent device geometry: the aperture width was set to $D_a = 8$ mm and each diffractive surface was 80 mm wide with 200×200 optimizable phase elements. In each corresponding figure, spatial dimensions are expressed in units of the center wavelength ($\lambda_2 = 0.75$ mm) to facilitate comparison with other figures. For this wavelength multiplexing scheme, we set $h_{max} = \max_i \frac{\lambda_i}{n(\lambda_i) - 1}$ to ensure that a full 2π phase modulation range is available at all the illumination wavelengths.”

7. In Fig. 10, why are the experimental figures in the first and fourth rows not rotated? According to the experimental protocol, they should be rotated if ϕ_{in} is not an integer multiple of 90° .

- We appreciate the reviewer's close attention to these details. We used two different holders, as shown in the figure below, to get different ϕ_{in} configurations. Measurements performed with Holder-2 resulted in the RFG being rotated by ± 15 degrees with respect to the horizontal; hence, the

corresponding patterns are rotated back by the same amount. Since the first and fourth row measurements were obtained with a holder with a horizontal base (Holder-1), they are not rotated.

8. In the discussion section, the authors mention the possibility of designing a unidirectional RFG that allows only forward transmission while blocking backward propagation. However, such functionality appears unfeasible, as it requires breaking time-reversal symmetry, which is typically achieved via magneto-optical effects. Diffractive surfaces, in general, do not violate time-reversal symmetry.

- This is an insightful comment raised by the reviewer. We have expanded the corresponding Discussion paragraph to explain the principle behind the design of **Unidirectional RFGs** and provided examples with **two newly added supplementary figures Figs. S6 and S7**:

*“...While time-reversal symmetry prohibits unidirectional behavior in a lossless electromagnetic system without e.g., magneto-optical effects, this constraint does not strictly apply to RFGs. Importantly, the RFG is not a lossless system: power is inherently lost at the edges of the finite-sized diffractive surfaces due to diffraction and material absorption. This allows for asymmetry in how loss manifests across the forward and backward propagation paths. By leveraging this asymmetry into our learning framework, we can engineer the loss distribution differently for the two directions, achieving functionally unidirectional behavior without violating reciprocity. To better highlight this, we define a composite loss function $L = L_{wf}^{(f)} + \eta_f(1 - DE^{(f)}) + \eta_b DE^{(b)}$, where L_{wf} defines the error between the output wavefront and the target wavefront (see Eq. 11), DE denotes the diffraction efficiency (see Eq. 13). The superscripts (f) and (b) denote the forward and backward directions, respectively, while η_f and η_b are training hyperparameters defining the weights of the respective loss terms. This formulation allows for promoting high diffractive efficiency in the desired forward direction while simultaneously suppressing efficiency in the undesired reverse direction, i.e., creating an asymmetric diffractive system. Simulation results for two RFG designs trained using this loss function are shown in **Supplementary Figs. S6 and S7**. In the first case, we set $\eta_f = 0$ and $\eta_b = 1$, only penalizing the diffraction efficiency in the backward direction. This results in approximately six orders of magnitude difference between the median diffraction efficiencies in the forward and backward directions, in favor of the forward direction, illustrating the unidirectional behavior of our design. As desired, the angular*

*error in the forward direction remains below 1°; see **Supplementary Fig. 6**. In the second case, reported in **Supplementary Fig. 7**, we simultaneously enforced high forward and low backward diffraction efficiencies by setting $\eta_f = 1$ and $\eta_b = 1$. This leads to further improvements in the forward diffraction efficiency (median 3.5%), while still maintaining strong suppression in the backward direction, with more than six orders of magnitude difference in median efficiencies between the forward and backward directions, once again showcasing the unidirectional behavior of our design. **These results demonstrate that asymmetric loss engineering during the supervised learning phase enables functionally unidirectional RFGs, despite being implemented using passive and reciprocal materials.***

9. Finally, I suggest adding a brief background discussion on the potential applications of programmable diffractive functions in various fields.

- We thank the reviewer for this valuable suggestion. Accordingly, we have expanded the following Discussion paragraph:

“...We believe that the capability to program refractive functions within passive materials could unlock unprecedented opportunities in manipulating optical waves, with major implications for optical device and system design across various applications. For example, RFGs could support applications in multi-channel optical interconnects, where distinct input directions can be routed to designated output channels with high spatial precision, enabling compact and scalable free-space optical communication links. Additionally, the intrinsic spatial and directional mapping abilities of RFGs can be harnessed for secure optical encoding, embedding information in complex, task-specific refractive transformations that are difficult to intercept or replicate without complete knowledge of the RFG design. Beyond these applications, RFGs could also be used for wave routing and optical switching. By replacing bulky optical components with compact, task-adaptive surfaces, these programmable diffractive platforms can significantly reshape photonic system design, driving advances in, e.g., communication, sensing, and imaging technologies.”

Reviewer #2:

The noteworthy results of this paper are:

1. the specific functionality presented: a negative refractive index volume, and wavelength-based electromagnetic multiplexing - the general claim being the implementation of the first "refractive function generator"
2. the engineering and design approach of the authors to implement a volume, passive, linear metamaterial in the THz regime using "supervised deep learning"

The major categories where this work seems to fit best are in electromagnetic inverse design and linear metamaterials.

The work is presented reasonably well with few clerical errors, and the breadth of the optimization of inputs is quite large. The mathematical framework underpinning the optimization is well elucidated, and the details of the optimization hyperparameters are provided. The experiment seems to agree relatively well with simulation qualitatively.

- We sincerely thank the reviewer for their positive and constructive evaluation.

However, this work to me does not seem to produce a significant scientific improvement in either of the fields. The design is done using a well understood electromagnetic model (Fourier optics), using a

gradient-based optimization method that the Ozcan group and others have previously presented. While the figure of merit seems to be somewhat different, the actual physics being simulated do not seem to be improved or changed at all. In addition, the performance of the devices themselves is relatively low, and the applicability of the devices to other adjacent fields also seems low.

Previous publications as an example (not exhaustive) by the Ozcan group include:

1. "all-optical neural networks..." DOI: 10.1126/science.aat8084
2. "computational imaging without a computer..." DOI: 10.1186/s43593-022-00012-4
3. "cascadeable all-optical NAND gates..." DOI: 10.1038/s41598-022-11331-4

- We thank the reviewer for this thoughtful perspective. While our wave propagation model is indeed based on well-established Fourier optics formalism, the scientific contribution of our work lies in the formulation and numerical/experimental validation of a learning-based framework for **refractive function generation (RFG)**. **Unlike prior works, including from our own group, that primarily targeted image classification or specific optical imaging or sensing-related functions, our approach in this work focuses on learning arbitrary angular transformations by training diffractive phase-only surfaces in an end-to-end fashion.** This recontextualizes diffractive design as a programmable routing problem in angular space, enabling new classes of wavefront control strategies that are not easily addressed by conventional inverse designs or physical intuition.

Furthermore, we demonstrate that complex trade-offs (e.g., between angular error and diffraction efficiency) can be explicitly engineered by modifying the learning loss function, which adds a layer of powerful design tunability absent from traditional approaches. Moreover, our paper presents concrete strategies for improving performance, including loss function engineering and 'vaccination' against fabrication misalignments or wavelength shifts. These techniques, combined with wavelength and polarization multiplexing demonstrated within the same thin volume, broaden the functional versatility of the platform. The generalizability to other domains, such as wave routing, optical encoding, and spatial multiplexing of channels, is also a possibility provided by our refractive function programming framework that distinguishes it from previous diffractive designs and opens up new system-level capabilities.

In summary, unlike earlier efforts that focused on diffractive optical networks for classification, logic operations, or direct computational imaging/sensing, this study aims to reframe how light refraction itself can be programmed. By proposing the arbitrary and independent control of output wave directions for each input direction, it bypasses the constraints of Snell's law and its generalized extensions. This reframing creates a new class of optical material behavior in the form of *refractive function generators* —a kind of angular I/O mapping that has not previously been demonstrated. **For example, our manuscript reports a distinct refractive function implemented at each wavelength through the same engineered material volume, i.e., the permutation of light refraction is switched from one desired function to another function by changing the illumination wavelength (see Fig. 7 and Supplementary Video 2).** In addition to the wavelength-multiplexed RFGs reported in our Results section, it is also possible to achieve **polarization multiplexing of refractive functions using isotropic transmissive materials constituting the RFG**; such an isotropic RFG design needs to be augmented with a separate, fixed polarizer array positioned between successive structured layers. **To demonstrate this capability, we designed an RFG with a fixed array of orthogonal polarizers placed after the fourth surface; see the newly added Supplementary Fig. S12.** The structure was trained to implement two distinct permutation refractive functions at two orthogonal linear polarization states, i.e., horizontal ($p=0^\circ$) and vertical ($p=90^\circ$). Simulation results show that the output angle errors are limited to $\sim 0.5^\circ$ across the entire input angular grid for both polarizations, confirming the feasibility of polarization-multiplexed refractive function generation using a shared diffractive volume. **The newly added Supplementary Video 4** also shows the far-field output intensity as the input wave direction is changed at these two

polarization states, together with the target patterns that follow the desired RFG at the corresponding polarization.

As another example, we reported the design of Unidirectional RFGs and demonstrated their performance with two newly added supplementary figures Figs. S6 and S7:

*“...While time-reversal symmetry prohibits unidirectional behavior in a lossless electromagnetic system without e.g., magneto-optical effects, this constraint does not strictly apply to RFGs. Importantly, the RFG is not a lossless system: power is inherently lost at the edges of the finite-sized diffractive surfaces due to diffraction and material absorption. This allows for asymmetry in how loss manifests across the forward and backward propagation paths. By leveraging this asymmetry into our learning framework, we can engineer the loss distribution differently for the two directions, achieving functionally unidirectional behavior without violating reciprocity. To better highlight this, we define a composite loss function $L = L_{wf}^{(f)} + \eta_f(1 - DE^{(f)}) + \eta_b DE^{(b)}$, where L_{wf} defines the error between the output wavefront and the target wavefront (see Eq. 11), DE denotes the diffraction efficiency (see Eq. 13). The superscripts (f) and (b) denote the forward and backward directions, respectively, while η_f and η_b are training hyperparameters defining the weights of the respective loss terms. This formulation allows for promoting high diffractive efficiency in the desired forward direction while simultaneously suppressing efficiency in the undesired reverse direction, i.e., creating an asymmetric diffractive system. Simulation results for two RFG designs trained using this loss function are shown in **Supplementary Figs. S6 and S7**. In the first case, we set $\eta_f = 0$ and $\eta_b = 1$, only penalizing the diffraction efficiency in the backward direction. This results in approximately six orders of magnitude difference between the median diffraction efficiencies in the forward and backward directions, in favor of the forward direction, illustrating the unidirectional behavior of our design. As desired, the angular error in the forward direction remains below 1° ; **see Supplementary Fig. 6**. In the second case, reported in **Supplementary Fig. 7**, we simultaneously enforced high forward and low backward diffraction efficiencies by setting $\eta_f = 1$ and $\eta_b = 1$. This leads to further improvements in the forward diffraction efficiency (median 3.5%), while still maintaining strong suppression in the backward direction, with more than six orders of magnitude difference in median efficiencies between the forward and backward directions, once again showcasing the unidirectional behavior of our design. **These results demonstrate that asymmetric loss engineering during the supervised learning phase enables functionally unidirectional RFGs, despite being implemented using passive and reciprocal materials.**”*

From a performance perspective, the maximum efficiency in the case of the negative refractive index material only reaches $< 60\%$ in simulation with low incident angle illumination. While there are many angles tested, this is still a discrete optimization, and there is significant symmetry in the negative refraction optical function. It seems there is some functional form in this case that the optimization is approaching imperfectly due to the non-convex nature of electromagnetic inverse design.

- We appreciate the reviewer’s valuable comment. To address this comment, we have added **a new supplementary figure Fig. S11** and the following Discussion paragraph in the revised manuscript:

*“...The RFG that was optimized for negative refractive function, shown in Fig. 6, was trained with a loss function that does not enforce any regularization related to diffraction efficiencies. As a result, the diffraction efficiencies were not uniform, with the high values only at lower angles of incidence. **We present in Supplementary Fig. S11 an alternative RFG design that emphasizes improved diffraction efficiency across a wider range of input directions.** The training loss function for this design was $L = L_{wf} + 5(1 - \theta_{in}/\theta_{max})(1 - DE)$, where L_{wf} is the error between the output wavefront and the target wavefront and DE is the diffraction efficiency (see Eqs. 11 and 13). For this new design, the diffraction efficiencies remain uniformly high over a wider range and fall to*

lower values only as θ_{in} approaches θ_{max} . This improvement in diffraction efficiency is achieved at the cost of a small increase in the angular error, albeit near the edges of the input angular range only, i.e., $\theta_{in} \approx \theta_{max}$. A detailed analysis of the output diffraction efficiency performance is presented in **Supplementary Fig. S11d**, where we quantified the angular power distribution at the output aperture. **At the bottom-right of Supplementary Fig. S11d, we report a ‘confusion matrix’ that shows the diffraction efficiencies along all the output directions \hat{k}_{out} for a given input direction \hat{k}_{in} . The dominantly flipped-diagonal structure of the confusion matrix (same as the structure of the matrix representing the negative refractive function, see Fig. 1) reveals the successful realization of the target refractive function.** However, for input directions at the edge ($\theta_{in} \approx \theta_{max}$), there is significant leakage of power out of the target direction at the output aperture. **The total output diffraction efficiency, obtained by summing over the rows of the ‘confusion matrix’, can approach 99.24%, as shown in the accompanying plot; see top-right of Supplementary Fig. S11d.”**

In the more general and challenging case of an 'arbitrary' angle mapping, the efficiency falls quite far to $< 5\%$. Contrary to the authors' insinuation that a 4-degree error with a 1% change in wavelength is good, it actually seems to be quite normal in the realm of linear metamaterials, but this is to be expected as the authors did not attempt to correct for chromatic aberration.

- We thank the reviewer for this comment. We have revised Fig. 3 and the following paragraph to address this comment:

“...In Fig. 3, we further analyze the errors of the refractive function implementation as the wavelength λ_{test} of the input light deviates from the design wavelength λ_{train} that the RFG is trained to operate at. While evaluating the error as a function of λ_{test} , we kept the input and target directions at λ_{test} the same as those at λ_{train} . At each test wavelength λ_{test} , the distribution of the angular errors (over N_m different input directions) is encapsulated with a box-and-whisker diagram in Fig. 3a. As expected, the error increases as λ_{test} deviates from λ_{train} . However, the angular errors remain below 4° over a wavelength range of $\sim 0.99\lambda_{train}$ to $\sim 1.01\lambda_{train}$. **The resilience against changes in wavelength can be improved by incorporating random wavelength variations during training. In Fig. 3b, we report the angular error of another design which was trained by randomly selecting the illumination wavelength from a desired range of interest during training, i.e., $\lambda_{train} \sim \text{Uniform}(742.5 \mu\text{m}, 757.5 \mu\text{m})$. The angular errors of this ‘vaccinated’ design remain limited to $\sim 1^\circ$ over the same wavelength range, showing the flexibility of our design approach to adapt to different requirements.”**

In some sense, the relatively low performance of the devices could be excused to some degree if the functionality seemed to demonstrate some application or use in the real world. The authors claim that "many-to-one" or "one-to-many" functionalities could be easily achieved without any demonstration. If it was so easily achieved, why is there no demonstration of this application? This would be great if true as it is definitely high impact and of fundamental interest to many beam combiners in AR/VR applications. These functions in general are thought to be quite difficult to achieve due to fundamental limitation of etendue, which apply to linear, passive systems.

- Following the reviewer's suggestion, we have added **new results on many-to-one mapping (see supplementary figure Fig. S10 and Supplementary Video 3)** and revised the following paragraph:

“Although the refractive functions that we discussed so far did not include many-to-one mappings between the input and output directions, i.e., more than one input directions do not result in the same output wave direction, this is not a restriction of the presented RFG framework. **It is possible to design RFGs that implement ‘many-to-one’ refractive functions and also one-to-many mappings, giving rise to input-direction-specific programmed beam-splitting. To**

demonstrate this capability, we designed an RFG that performs a many-to-one transformation, in which all the input directions of interest are mapped to the same output direction. As shown in Supplementary Fig. S10, the optimized RFG performs this mapping with low output angle error across the full input range. Supplementary Video 3 also shows the far-field output intensity as the input wave direction is swept, together with the corresponding target patterns that follow f ."

Other groups, notably the Faraon group have demonstrated similar functionalities that could be considered "refractive function generators" using inverse design in the mid-infrared range and simulated them in the visible range - notably focusing on functionalities which are relevant to the adjacent fields such as color routing for image sensors or wavefront sensing eg:

1. "Multifunctional volumetric meta-optics..." DOI: 10.1364/OPTICA.384228
2. "Multi-dimensional wavefront sensing..." DOI: 10.1364/OE.492440

The authors may also wish to consult a previous work which leverages this technique in an extremely low contrast system in the optical regime

1. "Aperiodic Volume Optics" DOI: 10.1038/nphoton.2009.290

- We thank the reviewer for pointing us to these references. We have enriched our reference section by appropriately adding them.

Finally, while there is significant detail in terms of the mathematics and hyperparameters, there does not seem to be any detail regarding the architecture of the tensorflow network used to implement the surfaces. It is customary in papers which utilize neural networks to disclose the architecture used in terms of layers used, activation functions, skips, pooling, etc. Most neural network papers have a figure which shows the architecture used in addition to a description.

- We appreciate the reviewer's valuable feedback. However, our framework does not utilize a neural network architecture in the conventional sense (e.g., layers, activations, or pooling). The learnable parameters correspond directly to the phase values at each pixel (half a wavelength in lateral dimensions) of the diffractive surfaces and are optimized using auto-differentiation and gradient descent (Adam) over a wave-based forward model using the angular spectrum approach. The wave-based forward model has been elaborately detailed in the Methods section to expand on these details.

Reviewer #3:

This paper introduces a compact refractive function generator that enables arbitrary light refraction by permuting the mapping between input and output directions, including multiplexed control via different wavelengths. The design is implemented using cascaded transmissive layers with optimized phase profiles and is experimentally demonstrated at terahertz frequencies. The work is innovative and opens new possibilities in static, function-specific optical control. While the work presents an innovative approach with promising potential in static, function-specific optical control, some expressions in the paper raise questions. My comments and questions are listed as follows

1. The term "programmable" is widely used in the research field of metasurface and integrated photonics; it typically implies dynamic, reconfigurable control at the level of individual elements. However, the proposed device has a fixed function once it is fabricated, which might not align with the

usual interpretation of "programmable." The authors are advised to clarify this terminology and consider a more accurate descriptor that better reflects the nature of the device.

- We appreciate the reviewer's insightful comments. In our work, the term "programming" refers to a one-time configuration process—analogue to writing data or a function onto a writable compact disc (CD)—rather than implying dynamic or real-time reconfigurability. Specifically, programming denotes the ability to encode arbitrary angular transformations or functions into the physical structure of the device through computational optimization. This is distinct from reconfigurable optical systems that permit post-fabrication tunability—more akin to re-writable CDs. While our current implementation employs passive transmissive surfaces, the same design framework could, in principle, be extended to dynamic architectures using spatial light modulators or tunable metasurfaces. To better represent our paper's focus, following the referee's suggestions, we have revised our title to: "*Programming of Refractive Functions*".

2. While wavelength-selective behavior is claimed in the paper, the device does not appear to function as a filter. Instead of absorbing or reflecting energy, the device redistributes the incident wave directionally. If the disappearance of a signal at a particular wavelength is due to spatial diffusion rather than reflection or absorption, the term "refractive function manipulation" may not be entirely appropriate. In other words, we cannot use the equivalent refractive index to calculate the energy distribution ratios between absorption, refraction, and reflection. The authors should clarify the physical mechanism behind this behavior to avoid any confusion.

- Our RFG devices redistribute the incident energy directionally through passive light-matter interactions that are programmed using deep learning for a desired function to be implemented. The mechanism is analogous to refraction in the broad sense: a deterministic change in propagation direction resulting from interaction with a structured medium. **Since our RFG structures do not support a well-defined bulk refractive index, we avoided the term "refraction" in isolation and instead used "refractive function" to emphasize the engineered angular transformations that we implement after supervised learning of the desired function.**

3. Based on the above two reasons, I believe that while the work is innovative, the title "Programmable refractive functions" is misleading. I suggest revising the title to better reflect the actual functionality.

- In response to the reviewer's suggestion, we have revised the title of our paper to "*Programming of Refractive Functions*", which more accurately reflects the scope and nature of the work. This is analogous to writing data or a function onto a writable compact disc (CD)—rather than implying dynamic or real-time reconfigurability. Our wording is also distinct from reconfigurable optical systems that permit post-fabrication tunability.

4. Since the authors aim to emphasize the editability of arbitrary output directions, it would be helpful to specify the time required to design and fabricate a device for a given refractive function. This information is crucial as it directly impacts the scalability and usability of the device in real-world scenarios.

- We have added the following to the Methods section to address the reviewer's comment:

"...The training time depends on several factors, including the number of diffractive surfaces and the

availability of GPU acceleration. For example, training a negative refractive function generator with $K = 5$ surfaces takes approximately 4 hours on an NVIDIA RTX 4090 GPU. The fabrication of a single diffractive surface takes approximately 1.5 hours on a Stratasys Objet30 V2 Pro 3D printer; the overall time can be significantly reduced by printing multiple surfaces simultaneously on the same build tray.”

5. The experimental results show relatively low diffraction efficiency, which is worth further analysis. The authors should investigate the underlying causes of this issue, such as phase quantization, fabrication errors, or design limitations. A discussion of possible improvements would enhance the clarity and completeness of the manuscript.

- Following the referee’s suggestions, we have added a **new supplementary figure Fig. S11**, together with the following Discussion paragraph to provide a detailed analysis of the diffraction efficiency of RFGs:

“...The RFG that was optimized for negative refractive function, shown in Fig. 6, was trained with a loss function that does not enforce any regularization related to diffraction efficiencies. As a result, the diffraction efficiencies were not uniform, with the high values only at lower angles of incidence. **We present in Supplementary Fig. S11 an alternative RFG design that emphasizes improved diffraction efficiency across a wider range of input directions.** The training loss function for this design was $L = L_{wf} + 5(1 - \theta_{in}/\theta_{max})(1 - DE)$, where L_{wf} is the error between the output wavefront and the target wavefront and DE is the diffraction efficiency (see Eqs. 11 and 13). **For this new design, the diffraction efficiencies remain uniformly high over a wider range and fall to lower values only as θ_{in} approaches θ_{max} .** This improvement in diffraction efficiency is achieved at the cost of a small increase in the angular error, albeit near the edges of the input angular range only, i.e., $\theta_{in} \approx \theta_{max}$. A detailed analysis of the output diffraction efficiency performance is presented in **Supplementary Fig. S11d**, where we quantified the angular power distribution at the output aperture. At the bottom-right of **Supplementary Fig. S11d**, we report a ‘confusion matrix’ that shows the diffraction efficiencies along all the output directions \hat{k}_{out} for a given input direction \hat{k}_{in} . **The dominantly flipped-diagonal structure of the confusion matrix (same as the structure of the matrix representing the negative refractive function, see Fig. 1) reveals the successful realization of the target refractive function.** However, for input directions at the edge ($\theta_{in} \approx \theta_{max}$), there is significant leakage of power out of the target direction at the output aperture. **The total output diffraction efficiency, obtained by summing over the rows of the ‘confusion matrix’, can approach 99.24%, as shown in the accompanying plot; see top-right of Supplementary Fig. S11d.**”

6. To strengthen the manuscript, it would be valuable to design a classical diffractive optical element, such as a focusing lens, using the proposed method. A comparison of its performance with traditional diffractive optical elements in terms of focusing efficiency, aberration, and compactness would provide useful context and highlight the advantages of the proposed approach.

- Following the reviewer’s suggestion, we have added a **new supplementary figure Fig. S8** and the following Discussion paragraph highlighting the advantages of our approach:

“...The supervised learning-based design approach we adopt offers several advantages. First, it naturally supports a ‘vaccination’ strategy, whereby the design is made resilient to anticipated

deviations from ideal conditions (such as wavelength shifts or hardware misalignments) by introducing such deviations as random noise during training; see Fig. 3b. Second, our framework allows for a desired trade-off among competing performance metrics to be explicitly controlled through loss function engineering, as demonstrated in Fig. 4. **To further highlight the versatility of our physics-based learning framework, we apply it to a classical task: design of light focusing elements.** While an ideal lens defined by the phase profile $\psi(x, y) = \frac{2\pi}{\lambda} \sqrt{x^2 + y^2 + f^2}$ can provide ideal focusing at wavelength λ and focal length f , its performance deteriorates when implemented using discretized phase elements. Under such constraints, our method can discover superior solutions. **Supplementary Fig. S8 presents designs of focusing optics comprising K diffractive surfaces, each consisting of phase elements discretized at $\lambda/2$ resolution and quantized into 8 uniformly spaced phase levels between 0 and 2π , i.e., 3 phase bit-depth.** For these simulations, we assume $\lambda = 400$ nm. The diameter of the diffractive surface(s) and the focal distance are assumed to be 288λ and 100λ , respectively, resulting in a numerical aperture of 0.82. The diameter of the focal spot (region of interest, ROI) is set to be λ . **To demonstrate the flexibility of tweaking the trade-off among performance metrics, we define two figures of merit relevant to energy localization: focusing efficiency (FE), which reports the ratio of optical power within the region of interest (ROI) to total input power, and power concentration ratio (PCR), which reports the ratio of power inside the ROI to that outside it.** As shown in Supplementary Fig. S8, the learned diffractive focusing designs outperform the ideal lens phase profile (phase-wrapped and quantized) in both FE and PCR, even when using the same design degrees of freedom with $K = 1$. With $K = 2$ diffractive surfaces, the performance improves further, demonstrating the advantages of structural depth in our framework. Moreover, the ability to explicitly tune the trade-off between FE and PCR via the loss function engineering further highlights the flexibility of our approach. This example illustrates how physics-based learning can push the boundaries of diffractive focusing beyond classical designs while remaining compatible with realistic fabrication constraints such as limited phase bit-depth.”